# Position: Beyond Sensitive Attributes, ML Fairness Should Quantify Structural Injustice via Social Determinants

**Zeyu Tang** [1]   **Alex John London** [2]   **Atoosa Kasirzadeh** [2]   **Sarah Stewart de Ramirez** [3]
**Peter Spirtes** [†2]   **Kun Zhang** [†24]   **Sanmi Koyejo** [†1]

## Abstract

Algorithmic fairness research has largely framed *unfairness as discrimination* along *sensitive attributes*. However, this approach limits visibility into *unfairness as structural injustice* instantiated through *social determinants*, which are contextual variables that shape attributes and outcomes without pertaining to specific individuals. **This position paper argues that the field should quantify structural injustice via social determinants, beyond sensitive attributes.** Drawing on cross-disciplinary insights, we argue that prevailing technical paradigms fail to adequately capture unfairness as structural injustice, because contexts are potentially treated as noise to be normalized rather than signal to be audited. We further demonstrate the practical urgency of this shift through a theoretical model of college admissions, a demographic study using U.S. census data, and a high-stakes domain application regarding breast cancer screening within an integrated U.S. healthcare system. Our results indicate that mitigation strategies centered solely on sensitive attributes can introduce new forms of structural injustice. We contend that auditing structural injustice through social determinants must precede mitigation, and call for new technical developments that move beyond sensitive-attribute-centered notions of fairness as non-discrimination.

## 1. Introduction

Structural injustice refers to circumstances in which social practices, social structures, or the environment reinforce

and compound prior histories of injustice (Carmichael et al., 1967; Sowell, 1972; Young, 1990; Tilly, 1998; Rothstein, 2017; Alexander, 2020). We use the term "social determinants" to denote the specific, measurable aspects of these practices, structures, or environments (i.e., the variables or features) through which structural injustice manifests and can be systematically audited (Gee & Ford, 2011; Yearby, 2018; Robinson et al., 2020; Yearby et al., 2022; Chetty et al., 2024). In contrast to auditing *unfairness as structural injustice* instantiated through social determinants, the machine learning (ML) fairness literature has largely operationalized auditing *unfairness as discrimination* along sensitive attributes, e.g., race, sex, gender, and age (Romei & Ruggieri, 2014; Loftus et al., 2018; Corbett-Davies & Goel, 2018; Mitchell et al., 2018; Narayanan, 2018; Verma & Rubin, 2018; Caton & Haas, 2020; Chouldechova & Roth, 2020; Makhlouf et al., 2020; Mehrabi et al., 2021; Zhang & Liu, 2021; Pessach & Shmueli, 2022; Tang et al., 2023). **In this position paper, we argue that beyond sensitive attributes, ML fairness should quantify structural injustice via social determinants.**

Because social determinants are features of places, institutions, policies, or practices, unfairness as structural injustice persists even if animuses that cause unjust treatment (unfairness as discrimination) have been subject to significant reform. Their effects may not be tied directly to demographic group membership but to broader traits (such as income level or job type) or to geographic areas. As a result, individuals within the **same** demographic group, depending on their unique circumstances, may experience **different** levels of (dis)advantage due to intersecting social determinants, e.g., various environmental impacts on health in terms of distance and accessibility to general practitioners and hospitals (Comber et al., 2011), Vitamin D and bone health across geographic locations (Yeum et al., 2016), and geographic disparities of diseases (Tan et al., 2020). Conversely, individuals from **different** demographic groups in the same geographic neighborhood may encounter **similar** impediments, e.g., poverty and pollution in the neighborhood, lack of educational resource in the community (Connell, 1994; Tilak, 2002; Rose & Dyer, 2008).

---

[†]Equal senior authorship. [1]Stanford University [2]Carnegie Mellon University [3]OSF HealthCare [4]Mohamed bin Zayed University of Artificial Intelligence. Correspondence to: Zeyu Tang <zeyu@cs.stanford.edu>.

*Proceedings of the 43rd International Conference on Machine Learning*, Seoul, South Korea. PMLR 306, 2026. Copyright 2026 by the author(s).

Auditing constitutes the essential first step in shaping the fairness problem space within which mitigation strategies operate. To support our position, our argument unfolds as follows. **(I) Conceptual Gap**: We identify the gap between cross-disciplinary treatments of social determinants for structural injustice, and their limited engagement in ML fairness research (Section 2). **(II) Paradigm Limits**: We show that the prevailing sensitive-attribute-centered technical paradigms are not well suited to quantifying structural injustice via social determinants (Section 3). **(III) Unintended Injustice**: We theoretically demonstrate that mitigation strategies focused on sensitive attributes can introduce additional structural injustice (Section 4). **(IV) Empirical Evidence**: We provide empirical evidence for the necessity and value of social-determinant-based analysis and intervention (Section 5). **(V) Actionable Reorientation**: We engage alternative perspectives (Section 6) and propose concrete actions for addressing the limitations of the current paradigm (Section 7).

## 2. Cross-Disciplinary Engagement with Social Determinants

In this section, we review and reflect on the cross-disciplinary engagement with social determinants.[1] In Section 2.1, we clarify definitions of sensitive attributes and social determinants. In Section 2.2, we summarize discussions on social determinants across disciplines.

### 2.1. Define Sensitive Attributes and Social Determinants

**Definition 2.1** (Sensitive Attributes). A *sensitive attribute* $A$, also referred to as a *protected feature* or a *social category*, is an intrinsic attribute of the individual that is canonically recognized in law, ethics, or social norms as warranting protection from discrimination or bias.

**Definition 2.2** (Social Determinants). A *social determinant* $S$ is a variable representing an aspect of the data generating process that satisfies:

(1) (Context-level definition) $S$ is defined at the level of a context (e.g., a neighborhood, institution, jurisdiction, or policy environment) rather than as an individual attribute. Multiple individuals share the same value of $S$ by virtue of being situated in the same context.

(2) (Social-structural content) $S$ characterizes a condition whose cross-context variation is substantially shaped by social-structural forces, such as resource allocation, institutional policy, or systematic investment. This encompasses both directly social conditions (e.g., school funding) and physical or environmental conditions that are socially patterned (e.g., pollution exposure).

(3) (Exogenous stratification) When $S$ is computed by aggregation over individuals, the grouping over which the aggregation is performed (e.g., a neighborhood boundary, jurisdiction, or institutional membership) is exogenously defined, not derived from the characteristics of the individuals being described.

Sensitive attributes are relatively stable identifiers arising from longstanding legal and moral frameworks, that can be uniquely ascribed to an individual, e.g., race, sex, disability status. Social determinants refer to external conditions that influence an individual's opportunities, behaviors, and outcomes (Carmichael et al., 1967; Sowell, 1972; Tilly, 1998; Singh, 2003; Young, 2008; Rothstein, 2017; Kind & Buckingham, 2018; Powers & Faden, 2019; Alexander, 2020; Kasirzadeh & Smart, 2021; Smart & Kasirzadeh, 2024; Sullivan & Kasirzadeh, 2024). Examples of social determinants include environmental impact on health, educational resource in the neighboring area, economic profile of the geolocation, collective values of the community, and so on.

To demonstrate the operationalization of distinguishing social determinants from sensitive attributes and their proxies, let us consider a simplified but realistic data generating process. Historical redlining channeled Black households into specific neighborhoods, entangling race, zip code, and neighborhood composition through a shared structural history (Rothstein, 2017). As a result, these variables remain strongly correlated over time. Despite this entanglement, Definition 2.2 separates them cleanly (Table 1).

Specifically, applicant's race is not a social determinant since it's not defined at the context-level. The zip code of an area is not a social determinant since it is an administrative label rather than a measure of any substantive condition. However, it is a proxy for social determinants, since it indexes real structural conditions (school funding, air quality, policing intensity) by partitioning space. This distinction matters for fairness implications: one cannot improve a zip code, but one can improve the school funding or air quality within it. The racial composition w.r.t. HOLC-redlined district and that w.r.t. zip code area differ in their grouping basis. HOLC redlining maps drew district boundaries explicitly based on the racial makeup of residents, making the aggregation endogenous to the characteristic being measured, whereas zip code boundaries are postal routes drawn independently of any demographic characteristic.[2]

### 2.2. Engagement Across Disciplines

In this subsection, we provide a high-level summary of discussions on social determinants from related disciplines of algorithmic fairness, including political philosophy, eco-

---

[1]Due to space limit, we provide further discussions on related works in Appendix A.

[2]We provide further discussion on social determinants and also their common presence in Appendix B.

*Table 1.* Demonstration of the operationalization of Definition 2.2.

| Variable | Context-level definition? | Social-structural content? | Exogenous stratification? | Categorization |
|---|---|---|---|---|
| Applicant's race | **No** | – | – | Sensitive attribute |
| Zip code | Yes | **No** | Yes | Not a social determinant |
| Racial composition (w.r.t. HOLC-redlined district) | Yes | Yes | **No** | Proxy for sensitive attribute |
| Racial composition (w.r.t. zip code area) | Yes | Yes | Yes | Social determinant |
| School funding per pupil (zip code area) | Yes | Yes | Yes | Social determinant |

nomics and sociology, and healthcare.

**Political Philosophy** In political philosophy, researchers have proposed to shift from a focus on distributive patterns to procedural issues of participation in deliberation and decision-making, and to consider structural injustices that arise from individuals' relations to contextual environments and social institutions (Blau, 1977; Giddens, 1979; Bourdieu, 1984; Young, 1990; 2006; 2008). Recent works in algorithmic fairness have urged a shift beyond the localized concerns of distributive justice, advocating for the need to investigate structural injustice (Kasirzadeh, 2022), and to explicitly incorporate procedural inquires into fairness assessments (Grgić-Hlača et al., 2018; Zimmermann & Lee-Stronach, 2022; Tang et al., 2024).

**Economics and Sociology** In the effort of quantitatively measuring social determinants, economists and sociologists have proposed various indices to capture the influence of contextual environments on individuals' opportunities and outcomes. For instance, the (updated) Area Deprivation Index (ADI) and Neighborhood Atlas are developed to rank neighborhoods by socio-economic disadvantage in a region of interest, e.g., at the state or national level (Kind et al., 2014; Kind & Buckingham, 2018). The Index of Concentration at the Extremes (ICE) measures spatial polarization of extreme privilege and deprivation (Massey, 2001; Kubrin & Stewart, 2006). The Child Opportunity Index (COI) measures the quality of resources and conditions that matter for children's healthy development in the neighborhoods where they live (Acevedo-Garcia et al., 2014).

**Healthcare** The social drivers of health (SDoH) have long been engaged in the healthcare literature. Researchers have pointed out that one size does not fit all when it comes to the index of socio-economic status in healthcare (Braveman et al., 2005), and that the incorporation of SDoH indices necessitates methodological clarity (Foryciarz et al., 2025). Previous works have found racial/ethnic and geographic variations in distrust of physicians in the U.S. (Armstrong et al., 2007). The distinction between healthcare costs and healthcare needs matters when it comes to the choice of target in the development of prediction algorithms (Obermeyer et al., 2019), and so does the distinction between race-based and race-conscious medicine (Pallok et al., 2019; Cerdeña

et al., 2020). Most recently, the World Health Organization (WHO) has released a report about the persisting social injustices (World Health Organization, 2025).

> **Takeaway:** Despite extensive cross-disciplinary work on structural injustice instantiated through social determinants, treatment within ML fairness remains limited.

## 3. Limitations of Current Technical Paradigms

In this section, we argue that current technical paradigms are inadequate for capturing unfairness as structural injustice.

### 3.1. Existing Practices and Benchmarks Tend to Drop Attributes Related to Social Determinants

In addition to individual-level variables, contextual environments have significant influences over the individual (Carmichael et al., 1967; Sowell, 1972; Tilly, 1998; Singh, 2003; Young, 2008; Rothstein, 2017; Kind & Buckingham, 2018; Powers & Faden, 2019; Alexander, 2020; Perdomo et al., 2025). For instance, for the variable Address (or its alternatives), the improvement in physical health was observed in a randomized housing mobility social experiment (Ludwig et al., 2011), and the social determinants of health are closely related to individual's residence area (Marmot & Wilkinson, 2005; Braveman & Gottlieb, 2014; Robinson et al., 2020; Yearby et al., 2022).

However, in ML fairness literature, it is a common practice to omit variables that do not directly pertain to individuals, when performing the prediction or decision-making tasks of interest. For instance, previous causal fairness approaches do not include address-related variables when modeling the data generating process with a causal graph (Kilbertus et al., 2017; Kusner et al., 2017; Nabi & Shpitser, 2018; Zhang & Bareinboim, 2018b;a; Chiappa, 2019; Wu et al., 2019; Imai & Jiang, 2020; Mishler et al., 2021; Coston et al., 2020; D'Amour et al., 2020; Nilforoshan et al., 2022; Nabi et al., 2022). Although the Communities and Crimes dataset (Redmond, 2009) initially contains geolocation, such information is dropped during data processing (Mary et al., 2019).

Moreover, widely used benchmark datasets typically exclude variables that potentially capture social determinants.

For instance, there is no address information included in the Adult dataset (Becker & Kohavi, 1996), which is a standard dataset for evaluation purposes (Calders et al., 2009; Zemel et al., 2013; Agarwal et al., 2018; Nabi & Shpitser, 2018; Donini et al., 2018; Agarwal et al., 2019; Baharlouei et al., 2020). Similar to the attribute list of Adult dataset, the address information is dropped by the Folktables package when retrieving public-use U.S. census data products to construct Adult-like prediction tasks (Ding et al., 2021), except for the ACSTravelTime task.

## 3.2. (Quasi-)Stable Sensitive Attributes Cannot Capture Shifting Influences from Social Determinants

In terms of the specification of the disadvantaged individuals, previous quantitative approaches in algorithmic fairness literature primarily focus on sensitive attributes. In addition to fairness notions that are applied one sensitive attribute at a time (Calders et al., 2009; Žliobaitė et al., 2011; Kamiran et al., 2013; Hardt et al., 2016; Zafar et al., 2017; Kilbertus et al., 2017; Kusner et al., 2017; Nabi & Shpitser, 2018; Chiappa, 2019), intersectional fairness considerations have been introduced to account for the intersection of multiple sensitive attributes through their structural combinations (Crenshaw, 1990; Bright et al., 2016; Kearns et al., 2018; Hoffmann, 2019; Foulds et al., 2020; Kong, 2022).

While a structured combination of multiple *(quasi-)stable* sensitive attributes provides a more nuanced characterization of intersecting factors in discrimination, it does not fully capture the *shifting* influence from contextual environments, in static and and dynamic settings. For instance, individuals with an identical configuration of sensitive attributes, e.g., the group of African American women, can face different levels of structural injustice depending on the contextual environments they are subjected to (Armstrong et al., 2007; Obermeyer et al., 2019; Pallok et al., 2019).

These influences are not intrinsic to the individual. Instead, they characterize the surrounding environment and dynamically shape the conditions under which outcomes are produced. Consequently, beyond asking which attributes explain (residual) dependence between individual's relatively stable sensitive attributes and outcomes (e.g., *Conditional Demographic Parity* by Žliobaitė et al. 2011; Kamiran et al. 2013 and further discussed by Wachter et al. 2021), fairness metrics should also adaptively and dynamically account for benefits/burdens induced by contextual variation such as neighborhood-level differences in healthcare access, exposure, and institutional resources.

## 3.3. Mismatch Between Individual-Level Causal Modeling and Community-Level Structure

In terms of the modeling of the instantiation of unfairness, previous causal fairness approaches represent discrimina-

tions with edges or pathways in the causal graph (Kilbertus et al., 2017; Kusner et al., 2017; Nabi & Shpitser, 2018; Chiappa, 2019; Wu et al., 2019; Coston et al., 2020; D'Amour et al., 2020; Nilforoshan et al., 2022; Nabi et al., 2022; Zuo et al., 2022), which typically originate from sensitive attributes of an individual. Social determinants, especially community- or context-level attributes (e.g., neighborhood deprivation or healthcare access), create a fundamental mismatch with most causal modeling frameworks in ML fairness, which primarily operate at the individual level.

Specifically, individual-level graph formulations do not readily represent community-level variables whose values emerge from shared environments and population composition. Many social determinants are aggregate statistics over populations, inducing potentially cyclic causal influence between individuals and communities. Individual-level causal interventions can propagate to the community level, and community conditions in turn shape individual outcomes. For example, intervening on an individual's income can shift neighborhood socio-economic status, affecting others in the community and violating standard assumptions such as *no interference* (Hernán & Robins, 2020) that underlie much of causal inference and fairness analysis.

Existing causal fairness notions further reinforce this individual-level structure by analyzing path-specific causal effects within individual-level graphs. Capturing effects of social determinants requires modeling additional structure across individuals and communities, which standard individual-level models do not encode. Although some work aggregates individual causal effects across subgroups (Coston et al., 2020; Imai & Jiang, 2020; Mishler et al., 2021), the underlying models remain individual-level, without capturing the bidirectional influences between an individual and their communities. Consequently, current causal fairness approaches are ill-suited to represent unfairness instantiated through collective, community-level social determinants.

> **Takeaway:** Existing technical paradigms, exemplified by standard benchmarks, sensitive-attribute-centered fairness metrics, and causal models of unfairness as discrimination, are inadequate for capturing structural injustice.

## 4. Tension Between Structural Justice and Sensitive-Attribute-Centered Mitigation

To formalize the intuition that identity-based metrics fail to capture structural context, we introduce a stylized model and analyze college admissions as an instance of sensitive-attribute-centered mitigation grounded in U.S. affirmative-action efforts to remedy race-based historical injustice. We theoretically demonstrate that incorporating social determinants, even through a simple geographic proxy, not only recovers nuances consistent with prior qualitative discus-

sions, but also uncovers a counter-intuitive paradox between sensitive-attribute-centered mitigation and progress in structural justice. In Section 4.1, we present assumptions we use to facilitate closed-formula theoretical analyses. In Sections 4.2, we demonstrate structural injustice implications of a classic sensitive-attribute-centered mitigation strategy.[3]

## 4.1. Assumptions in Our Analyses

**Assumption 4.1** (Region-Specific Demographic Makeup). Let us denote the sensitive attribute as $A$, where $a \in \mathcal{A}$ denotes under-represented minority (URM) applicant group, and $a' \in \mathcal{A}$ denotes non-URM applicant group. There are two regions where applicants reside in, rich and poor regions, with different demographic compositions,

|  | poor region | rich region |
|---|---|---|
| URM applicants | $n_a^{(\text{poor})}$ | $n_a^{(\text{rich})}$ |
| Non-URM applicants | $n_{a'}^{(\text{poor})}$ | $n_{a'}^{(\text{rich})}$ |

where the following inequalities hold true:

(1) geographic disproportion due to historical injustice, i.e., $n_a^{(\text{poor})}/n_{a'}^{(\text{poor})} > n_a^{(\text{rich})}/n_{a'}^{(\text{rich})}$,

(2) the definition of "underrepresented minority," i.e., $n_a^{(\text{poor})} + n_a^{(\text{rich})} < n_{a'}^{(\text{poor})} + n_{a'}^{(\text{rich})}$.

Condition (1) specifies that URM applicants are relatively more concentrated in the less well-off region due to historical injustice (Sowell, 2004; Rothstein, 2017; Alexander, 2020). Condition (2) holds by definition, i.e., the total number of URM applicants is smaller than that for non-URM applicants.

**Assumption 4.2** (Determinant of Academic Preparedness). Conditioning on the affluence of the region where the applicant resides in, the academic preparedness is conditionally independent from the sensitive attribute race. In other words, we have the following relation ($\perp\!\!\!\perp$ denotes independence):

`Academic Preparedness` $\perp\!\!\!\perp$ `Race | Region.`

While there can be marginal dependence between `Race` and `Academic Preparedness` due to historical injustice (Sowell, 2004; Rothstein, 2017; Alexander, 2020), such dependence does *not* indicate that `Race` is a determinant of applicant's `Academic Preparedness`. Assumption 4.2 specifies that after conditioning on applicant's address region (and therefore, region-specific social determinants related to education), applicant's academic preparedness is irrelevant to the demographic group. This assumption is not intended to use region as a mediating variable to explain racial discrimination in education, e.g., as in *Conditional Demographic Parity* frameworks (Žliobaite et al.,

---

[3]Additional results and proofs are provided in Appendix C.

2011; Kamiran et al., 2013; Wachter et al., 2021). Instead, it encodes the premise that race is not an inherent factor shaping learning ability, explicitly refuting essentialist interpretations and aligning with the literature (Roberts, 2011; Kohler-Hausmann, 2018; Smedley, 2018; Kasirzadeh & Smart, 2021; Delgado & Stefancic, 2023; Doh et al., 2025).

For tractability, Assumption 4.2 simplifies the underlying data generating processes by focusing on inter-regional structural injustice and abstracting from intra-regional and within-institution racial discrimination (e.g., bias operating within the same school), which remain important mechanisms beyond the scope of our model. Even in absence of these mechanisms, we show that the interplay between sensitive-attribute-centered mitigation and progress in structural justice remains non-trivial.

**Assumption 4.3** (Stochastic Dominance of Academic Preparedness Distribution). Let $S$ denote the non-negative overall academic index score of an applicant's academic preparedness. Further let $S_{\text{MAX}}$ and $S_{\text{MIN}}$ denote the highest and lowest possible values of the score. The rich region's cumulative distribution function (CDF), of log-converted relative score $Q$ dominates that of the poor region:

Let $Q := -\log\left(\dfrac{S - S_{\text{MIN}}}{S_{\text{MAX}} - S_{\text{MIN}}}\right)$, then for $r \in \{\text{poor}, \text{rich}\}$,

the CDF's satisfy $F^{(\text{rich})}(q) \geq F^{(\text{poor})}(q), \forall q \in [0, \infty)$.

The degree of stochastic dominance in academic preparedness between applicants from rich and poor regions captures the level of structural injustice.

**Assumption 4.4** (Selective Admission and Open Enrollment). The selective college employs thresholds on applicants' academic preparedness scores and has a limited availability of admissions $g$:

$g < n$, where $n = n_a^{(\text{poor})} + n_a^{(\text{rich})} + n_{a'}^{(\text{poor})} + n_{a'}^{(\text{rich})}$,

all applicants can get admitted to open-enrollment college.

## 4.2. Structural Injustice Implications of A Classic Mitigation Strategy

The quota-based admission is a type of affirmative-action admission strategy, originally designed to rectify historical injustice by directly setting aside admission quotas to increase the representation of URM students. Aside from the fact that the quota-based admission procedure is rigid and mechanical (*Grutter v. Bollinger (2003)* Supreme Court, 2003b), this sensitive-attribute-centered mitigation strategy fails to account for the role of social determinants, which vary across regions and influence academic preparedness.

Our results quantitatively recover the well-known concern that quota-based admissions reduce opportunities for non-URM students from disadvantaged regions by shrinking

the pool of open seats. Moreover, we uncover a **counter-intuitive paradox**: this unintended burden is less likely under more severe structural injustice, and more likely as conditions improve.

**Theorem 4.5** (Tension Between Structural Justice and Sensitive-Attribute-Centered Mitigation). *Under Assumptions 4.1–4.4, let us denote with $\eta_{\text{quota}} \in \left[1, \frac{n}{n_a^{(\text{poor})}+n_a^{(\text{rich})}}\right]$ the weighting coefficient over the natural proportion of URM applicants in population, such that the quota for URM admissions in the selective college is $\eta_{\text{quota}} \cdot (\frac{n_a^{(\text{poor})}+n_a^{(\text{rich})}}{n}g)$. Then, the quota-based admission strategy imposes a more competitive requirements (in terms of score threshold) for non-URM applicants from the poor region, than that for URM applicants from the rich region, **unless** the following condition is satisfied:*

$$\max_{q \in [0, \infty)} \frac{F^{(\text{rich})}(q)}{F^{(\text{poor})}(q)} \geq \frac{\eta_{\text{quota}}}{1 + (1 - \eta_{\text{quota}}) \frac{n_a^{(\text{poor})}+n_a^{(\text{rich})}}{n_{a'}^{(\text{poor})}+n_{a'}^{(\text{rich})}}} \quad . \quad (1)$$

In simple terms, Theorem 4.5 establishes that (i) quota-based mitigation can disadvantage non-URM applicants from disadvantaged regions, and (ii) counter-intuitively, such harm is easier to avoid when structural injustice is more severe, and more aggressive sensitive-attribute-centered mitigation only makes generating new injustice more likely.

Specifically, as the quota increases ($\eta_{\text{quota}}$ grows), more seats are reserved for URM applicants (from both poor and rich regions), the more challenging for non-URM applicants in the poor region to be able to attend the selective college. Non-URM applicants in the poor region, who face the same obstacles and disadvantages in contextual environments as their URM counterparts, are not reserved additional spots; on top of that, they have to compete with more advantaged peers (non-URM applicants from the rich region) for a shrinking pool of openings.

More importantly, and counter-intuitively, this unintended harm is less likely when existing structural injustice is more severe, as captured by a larger degree of stochastic dominance on the left-hand side of Equation (1). However, as structural justice improves and regional dominance diminishes, quota-based mitigation becomes increasingly likely to generate new forms of structural injustice, since the inequality in Equation (1) can get violated. Moreover, the right-hand side of Equation (1) grows with the quota size $\eta_{\text{quota}}$, indicating that stronger sensitive-attribute-centered intervention amplifies this unintended harm more drastically.

Unlike traditional arguments against affirmative action that seek to ignore certain sensitive attributes, we argue that ignoring the interaction between sensitive attributes and social determinants harms the most disadvantaged subsets of all demographic groups.

> **Takeaway:** There is a paradoxical relationship between sensitive-attribute-centered mitigation and structural injustice: certain mitigation strategies are less likely to introduce new disadvantage when existing structural injustice is more severe, but increasingly generate new forms of structural injustice as conditions improve.

## 5. Empirical Analyses

In this section, we provide empirical analyses of real-world interplay between sensitive attributes and social determinants. In Section 5.1, we demonstrate the insufficiency of (intersectional) sensitive attributes when capturing disadvantage on the U.S. Census data (Census Bureau, 2009; 2014; 2022). In Section 5.2, we present a semi-synthetic analysis in a high-stakes domain application regarding breast cancer screening within an integrated U.S. healthcare system.

### 5.1. Enhancing Census Data Product Though Linking Socio-Economic Status Indices

We retrieve the public use microdata sample (PUMS) data from the U.S. Census Bureau (Census Bureau, 2023), and provide visualizations of different Public Use Microdata Areas (PUMAs) in California based on the 2023 U.S. Census PUMS data. Each PUMA contains at least $100,000$ residents and provides reliable, detailed demographic, economic, and housing statistics at a sub-state level while also protecting the confidentiality of respondents. In order to link PUMAs to indicators of (area-level) social determinants, we also retrieve the updated Area Deprivation Index (ADI).[4]

In Figure 1, we present the histogram of annual income for African American women residing in PUMAs with different ADI levels. As we can see, although the demographic information reflects the intersectional characteristics of individuals (race and sex), the social determinants in different regions still play a nontrivial role in shaping the income distribution. For instance, in PUMAs with higher ADI levels, the income distribution is more skewed towards lower income levels, indicating that individuals in these areas may face more significant economic challenges compared to those in PUMAs with lower ADI levels. We provide further analyses on PUMAs in Appendix D.

### 5.2. Semi-Synthetic Analysis in a High-Stakes Domain: Breast Cancer Care

In addition to the demographic analyses based on the U.S. census data, we also analyze de-identified patient records from OSF HealthCare, an integrated healthcare system spanning multiple hospitals and outpatient facilities

---

[4]https://www.neighborhoodatlas.medicine.wisc.edu/

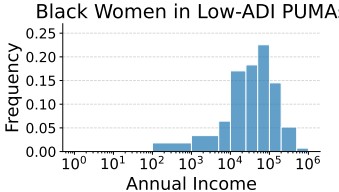 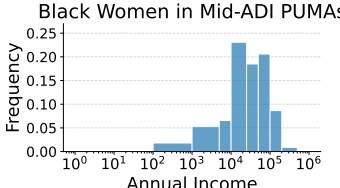 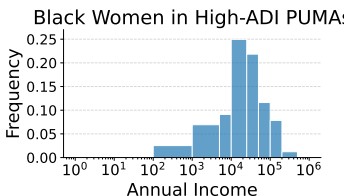

*(a)* PUMAs w/ low-level area deprivation     *(b)* PUMAs w/ mid-level area deprivation     *(c)* PUMAs w/ high-level area deprivation

*Figure 1.* Histogram of annual income for African American women residing in different areas. Sensitive attributes (race and sex) are shared across the subfigures, whereas social determinants are not (higher ADI indicates higher area deprivation). The median annual income for African American women is $38,000 for low-ADI PUMAs, $23,800 for med-ADI PUMAs, and $18,800 for high-ADI PUMAs.

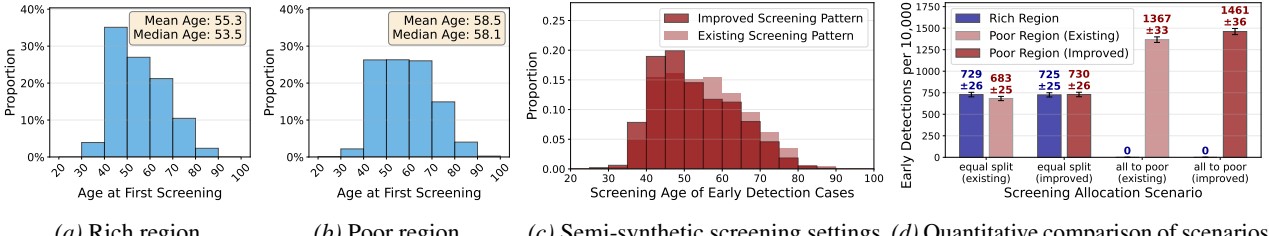

*(a)* Rich region     *(b)* Poor region     *(c)* Semi-synthetic screening settings     *(d)* Quantitative comparison of scenarios

*Figure 2.* Panels (a) and (b): Histogram of the age at the first-ever breast cancer screening for White women residing in different areas. Sensitive attributes (race and sex) are shared, whereas social determinants are not: the rich region corresponds to $\text{ADI} \in [0, 25)$ and the poor region corresponds to $\text{ADI} \in [75, 100]$. Panels (c) and (d): We conduct a semi-synthetic simulation of breast cancer onset and screening to evaluate how policy interventions targeting social determinants affect early detection outcomes. Panel (c) shows that adopting improved (rich-region) screening pattern in the poor region shifts detections earlier. Panel (d) demonstrates quantitatively that such structural improvements yield non-trivial gains in early detections across allocation scenarios (with a same total number of screenings).

across a multi-county region, serving both urban and rural communities. Focusing on breast cancer screening and care from 2012–2022, we linked cancer registry-derived mammography logistics data (e.g., screening and diagnostic mammography dates and locations) with demographic and clinical variables (e.g., birth date, gender, race/ethnicity, residential ZIP code, residential area socio-economic index values) using anonymized patient identifiers to ensure full de-identification. The final dataset contains approximately 54,000 screening mammography records from around 45,000 unique patients.

In Figures 2(a) and 2(b), we present the distribution of age at first breast cancer screening for White women across regions within the healthcare system. As we can see, even among individuals with identical sensitive attributes (White women), the age at first-ever breast cancer screening differs by several years (e.g., over 3 years in the mean and nearly 5 years in the median). Because the integrated healthcare system applies a uniform screening guideline across regions, these gaps cannot be attributed to region-specific recommendations.[5] Instead, they are consistent with structural disadvantages, such as differences in transportation infras-

tructure, access to care, or other contextual barriers, that shape screening uptake. The fact that the guideline is not explicitly social-determinant-aware indicates the inadequacy of fairness through unawareness, mirroring prior insights regarding sensitive attributes (Dwork et al., 2012; Lipton et al., 2018), now situated in the context of social determinants.

To illustrate the impact of policy interventions targeting social determinants, we perform a semi-synthetic simulation. We use 100,000 particles (the minimum PUMA population size) per region (rich and poor) with ages uniformly distributed from 20-99. For each particle in both regions, we simulate whether and when breast cancer develops by sampling year-by-year from age 20 onwards using SEER age-specific incidence rates (2018-2022)[6], so that the simulated age-specific onset rate is consistent with epidemiological data. We then allocate 10,000 screening slots across regions according to different policy scenarios. For each screening, we sample a "first-ever screening age" from empirical distributions derived from real data. An "early detection" benefit occurs when a particle is screened at or before her cancer onset age (assuming reliable periodic screenings af-

---

[5]Nevertheless, across the United States, breast cancer screening guidelines broadly agree that screening mammography should begin around age 40 for average-risk individuals, but differ on recommended frequency.

[6]https://seer.cancer.gov/statistics-network/explorer/application.html?site=55&data_type=1&graph_type=3&compareBy=race&chk_race_1=1&rate_type=2&sex=3&advopt_precision=1&advopt_show_ci=on

ter the initial screening), representing cases where screening enables earlier breast cancer care.

Figure 2(c) shows screening ages at which early detections occur. The "existing screening pattern" follows the poor region's empirical distribution of age at first-ever screening (Figure 2(b)), and the "improved screening pattern" follows that of the rich region (Figure 2(a)). Both screening patterns allocate all 10,000 screenings to the poor region, differing only in the first-screening age distribution used. From the overlaid histograms, we can see that improvements in social determinants, instantiated through improved screening patterns in the poor region, lead to earlier detection timing.

Figure 2(d) compares four allocation scenarios for 10,000 new screenings (500 simulation runs): equal split (5,000 each region) versus all to the poor region, crossed with existing versus improved screening patterns. Shifting to an improved screening pattern yields a nontrivial gain, for example, increasing early detections from $1,367 \pm 33$ to $1,461 \pm 36$ per 10,000 screenings allocated to the poor region.[7]

> **Takeaway:** We empirically demonstrate the instantiation of social determinants beyond sensitive attributes and, via semi-synthetic experiments, show the nontrivial benefits of policies that improve them.

## 6. Alternative Views

### 6.1. Viewpoint: Social Determinants Are "Just Another" Sensitive Attribute from a Technical Perspective

This alternative view states that, technically, social determinants can be treated as "just another" sensitive attribute, and therefore, there is no need to capture structural injustice specifically through social determinants:

> *Under associative fairness metrics, the influence of social determinants is instantiated as distribution shifts across domains, and can be addressed through domain adaptation mechanisms.*

We respond to this alternative view by arguing that social determinants are not inherently individual-level attributes that merely shift across domains. Rather, they reflect structural injustices that shape individuals opportunities and outcomes in ways that are detached from specific identities or demographic group memberships. Domain adaptation and transfer learning style fairness approaches typically ask (Madras et al., 2018; Coston et al., 2019; Schumann et al.,

2019; Creager et al., 2021; Mukherjee et al., 2022; Teo et al., 2023): "How can we learn predictors that eliminate empirical measures of discrimination under distribution shifts over observed individual attributes, so that disparities across individuals or demographic groups is robustly mitigated?" This framing treats heterogeneity across domains as variations in observable distributions, while keeping discrimination defined relative to individual/group identity.

In contrast, inquiries centered on social determinants ask a fundamentally different question: "How can we identify and quantify latent structural factors that give rise to these domains in the first place, i.e., factors whose (dis)advantageous effects on individuals are not tightly bound to individual identities or demographic categories?" Essentially, the goal is to capture both **within-group** heterogeneity (different levels of advantage among individuals of the same demographic group) and **cross-group** structural burdens (similar disadvantages experienced across different demographic groups). Social determinants operate at a structural level that is not intended to be eliminated through distributional robustness of a predictor, and thus cannot be treated as merely another sensitive attribute. It is also important to distinguish our position from redlining (Rothstein, 2017): rather than using correlated non-sensitive attributes (e.g., social determinants) as proxies for sensitive attributes to facilitate discrimination, we seek to audit how structural injustice is instantiated so it can be systematically addressed.

### 6.2. Viewpoint: Causal Effects of Sensitive Attributes Already Encompass Social Determinants

This alternative view states that, the influence of social determinants are captured either implicitly or explicitly by the causal effects of sensitive attributes:

> *For causal fairness metrics, social determinants are modeled either implicitly (encoded in directed edges from sensitive attributes to downstream variables) or explicitly (as mediators along the causal pathways originating from sensitive attributes), and therefore, their influences are encompassed in causal effects of sensitive attributes.*

We respond by arguing that causal effects of sensitive attributes do not correspond to interventions on the underlying structural conditions, and therefore, cannot recover the causal influence from social determinants.

First, for the "implicit" case, representing social determinants through directed causal edges from sensitive attributes to downstream variables creates a mismatch between the technical meaning of the causal model and the fairness objectives it is intended to capture. Let us consider Race → Education Status (Kusner et al., 2017; Nabi & Shpitser, 2018; Chiappa, 2019; Nabi et al., 2022) as an example. Even if we treat directed edges only as reductive summaries (e.g., encoding the influence

---

[7]We acknowledge that our semi-synthetic simulation abstracts from the full complexity of underlying epidemiological processes. Beyond first-time screening, incidence is also shaped by additional factors, e.g., genetic (Collins & Politopoulos, 2011; Shiovitz & Korde, 2015; Pal et al., 2024), epigenetic (Kim et al., 2023; Prabhu et al., 2024), that may shift observed onset. Therefore, we do not intend to make, nor should our results be interpreted as, causal claims about structural barriers.

of unobserved factors such as neighborhood affluence, geographic location, or institutional access) instead of asserting direct causal effect, there is a mismatch: while the edge stands in for latent contextual influences, the contrast operates on sensitive attributes themselves rather than on the underlying social determinants it proxies.

Second, for the "explicit" case, while previous path-specific fairness approaches (Zhang & Bareinboim, 2018a; Chiappa, 2019; Wu et al., 2019) can potentially include social-determinant variables as mediators along causal pathways from sensitive attributes to outcomes, this framing treats such variables as static downstream features rather than as shifting structural levers that could be the target of intervention. Moreover, social determinants need not be downstream variables of sensitive attributes. For example, race or sex is not a plausible ancestor node of environmental or contextual variables, which affect individuals in ways that are not mediated by, nor specific to, demographic memberships.

> **Takeaway:** Both alternative views collapse social determinants into sensitive attributes, but doing so obscures their fundamentally non-interchangeable conceptual roles and leads to misaligned technical abstractions.

## 7. Call to Actions and Concluding Remarks

Addressing structural injustice in ML fairness demands coordinated advances in data governance, dynamic metric design, and explicit characterization of underlying data generating processes. In this section, we outline these actionable pillars.

**Pillar 1 Data Governance: Privacy-Aware Use of Social Determinant Data**  Institutions should establish tiered data governance frameworks that distinguish between raw identifiers, privacy-preserving aggregates, and analytically derived contextual features. Concretely, raw identifiers (e.g., exact address/community, census tract) should be stored in restricted data locations with audited access, while model development relies on privacy-preserving contextual variables such as neighborhood deprivation indices, environmental exposure measures, or resource accessibility scores computed via differential privacy (Dwork et al., 2014) mechanisms. Data curators should document how social determinant variables are constructed, quantify privacy risk, and provide standardized contextual feature sets that can be safely integrated into datasets and benchmarks.

**Pillar 2 Metric Design: Dynamic and Context-Adaptable Fairness Metrics**  Fairness metrics should be redesigned to track disparities across evolving structural conditions rather than static demographic groupings or individual identities alone. Practically, this involves coupling individual-level attributes with time-varying contextual indicators (e.g., level of environmental pollution, policy exposure, access to care) and computing disparity measures conditional on these structural states. For example, as a counterpart to *Demographic Parity* (Calders et al., 2009; Dwork et al., 2012; Zemel et al., 2013; Feldman et al., 2015), *Social Determinant Parity* could evaluate (predicted) outcome disparities across area deprivation indices, infrastructure access levels, or regions with differing policy exposure. In addition, longitudinal formulations of *Social Determinant Parity* can track how outcome disparities change as contextual variables shift over time, enabling ongoing diagnostic of how algorithmic systems interact with changing social environments.

**Pillar 3 Underlying Data Generating Processes: Identifying Intervention Levers**  Causal fairness frameworks should treat social determinants, especially community- and context-level conditions, as explicit intervention targets. Concretely, causal fairness approaches should construct multi-layer causal models in which individual outcomes are jointly influenced by personal attributes and contextual variables such as neighborhood resources, environmental exposures, and policy exposures. This shift also necessitates leveraging causal representation learning (Xie et al., 2020; Schölkopf et al., 2021; Zheng & Zhang, 2023; Xie et al., 2024; Ng et al., 2024; Zhang et al., 2024; Dai et al., 2025) to actively seek out latent factors that could reflect structural characteristics, as well as hybrid inference methods that jointly perform causal inference (Pearl, 2009; Peters et al., 2017; Hernán & Robins, 2020) and relational inference between individuals and their surrounding contexts (Friedman et al., 1999; Maier et al., 2010; Ahsan et al., 2023).

> **Conclusion:** Addressing structural injustice requires ML fairness research to provide technical tools that extend beyond addressing sensitive attributes, enabling the identification of effective structural levers and the rigorous measurement of progress toward structural justice, so that policy interventions can be better informed and evaluated.

## Acknowledgments

We would like to acknowledge the support from NSF Award No. 2229881, AI Institute for Societal Decision Making (AI-SDM), the National Institutes of Health (NIH) under Contract R01HL159805, and grants from Quris AI, Florin Court Capital, MBZUAI-WIS Joint Program, and the Al Deira Causal Education project. SK acknowledges support by NSF 2046795 and 2205329, IES R305C240046, ARPA-H, the MacArthur Foundation, Schmidt Sciences, and HAI. We appreciate constructive feedback from anonymous reviewers, and we gratefully acknowledge Clark Glymour, Safura Sultana, Jonathan Handler, William Bond, Marlene Robles Granda, David McGrew, Corey Silver, Kelly Johnson, and George Sanders for their valuable project support and assistance.

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

# Supplement to "Position: Beyond Sensitive Attributes, ML Fairness Should Quantify Structural Injustice via Social Determinants"

## Table of Contents: Appendix

# A. Further Discussions on Related Works

In this section, we present further discussions on related works. In Section A.1, we consider types of information utilized when characterizing algorithmic fairness, and their relative emphases. In Section A.2, we provide a detailed comparison between our advocacy and previous works on causal fairness. In Section A.3, we present additional remarks including the use of term "structure" and the conceptual distinction between "unfairness" and "discrimination" in related disciplines.

## A.1. Fairness Notions Based on Observational Statistics and Causal Analysis

Various notions have been proposed in the algorithmic fairness literature to characterize fairness with respect to the prediction or the prediction-based decision-making (Dwork et al., 2012; Hardt et al., 2016; Chouldechova, 2017; Zafar et al., 2017), and also notions that are based on causal modeling of the data generating process (Kusner et al., 2017; Kilbertus et al., 2017; Nabi & Shpitser, 2018; Chiappa, 2019; Wu et al., 2019; Coston et al., 2020). Recent survey papers have presented overviews on fairness notions in static settings (Loftus et al., 2018; Makhlouf et al., 2020; Mehrabi et al., 2021), dynamic settings (Zhang & Liu, 2021), and also the connection between algorithmic fairness and the literature from moral and political philosophy (Tang et al., 2023).

The type of information utilized reflects different emphases of algorithmic fairness studies. Notions based on observational statistics analyze the fairness implications in terms of the *outcome* of predictions or decision-making (Dwork et al., 2012; Hardt et al., 2016; Chouldechova, 2017; Zafar et al., 2017; Kearns et al., 2018; Foulds et al., 2020). Approaches that capture causal influences from the protected feature to the target variable at the individual-level (Kusner et al., 2017; Kilbertus et al., 2017; Nabi & Shpitser, 2018; Chiappa, 2019; Wu et al., 2019) and the (sub-)group-level (Coston et al., 2020; Imai & Jiang, 2020; Mishler et al., 2021) put more emphases on the *procedural* aspect of algorithmic fairness inquiries, focusing on the data generating process of interest. Recent work has also proposed to address procedural fairness over all objectionable data generating components (Tang et al., 2024) according to John Rawls's advocacy for pure procedural justice (Rawls, 1971; 2001).

## A.2. Detailed Comparison with Causal Fairness Approaches

In this subsection, we provide a detailed comparison between our position and previous works on causal fairness, in terms of the question of interest (Section A.2.1), and whether or not our framework are in tension with previous causal fairness approaches (Section A.2.2).

### A.2.1. QUESTION OF INTEREST

To avoid overloading the term "counterfactual" in the causal inference literature (Spirtes et al., 1993; Pearl, 2000; Peters et al., 2017), we use "counter-factual" (with a hyphen, as an opposite to "factual") to denote that something does not happen in the current reality. Previous causal fairness approaches have utilized interventional (Kilbertus et al., 2017; Nabi & Shpitser, 2018; Nabi et al., 2019; 2022) and/or counterfactual (Kusner et al., 2017; Chiappa, 2019; Wu et al., 2019) causal effects in the technical formulation, and aim to answer the following question:

**Question A.1** (**Counter-Factual Analysis Starting from Protected Features**). Under certain conditions and assumptions, what would happen to the predicted outcome in the factual world and the counter-factual world, had **the protected feature(s)** taken different values?

Based on estimating or bounding certain causal effects among variables, including the protected feature, the (predicted) outcome, and certain variables that are closely related to but not the protected feature itself, e.g., proxy variables (Kilbertus et al., 2017), redlining attributes (Zhang et al., 2017), admissible variables (Salimi et al., 2019), and so on, the fairness violation is quantified in terms of causal effects between the protected feature and the (predicted) outcome. There is a reductive focus solely upon the protected feature when modeling the discrimination. For instance, it is a common practice for causal fairness notions to consider varying the value of protected feature (Kilbertus et al., 2017; Kusner et al., 2017; Nabi & Shpitser, 2018; Nabi et al., 2019; 2022; Chiappa, 2019; Wu et al., 2019) as the starting point. Recently, Tang et al. (2024) have also proposed to consider not only edges or paths originating from the protected feature, but also all objectionable components in the data generating process, to address procedural fairness.

However, the modeling choice of "summarizing" discrimination only through edges/paths originating from protected feature, or solely among individual-level variables, falls short of the need to capture structural injustice. The characteristics of the

environment and the context that individuals operate in typically do not correspond to individual-level attributes, and are not considered in previous literature. Different from causal fairness approaches, our position calls for explicit incorporations of the influence of contextual environments, and aims to address questions like:

**Question A.2** (**Factual Analysis Incorporating Social Determinants**). Under certain conditions and assumptions, what are the aspects of the data generating process that characterize **the influence from contextual environments to the individual**?

While social determinants often correlate with sensitive attributes, they cannot be captured by features of any particular individual. Explicit consideration and modeling of social determinants facilitate a more comprehensive understanding of the benefits and burdens experienced by individuals from diverse demographic backgrounds as well as contextual environments, which is essential for understanding and achieving fairness effectively and transparently.

### A.2.2. NO CONFLICT IN PRINCIPLE WITH CAUSAL FAIRNESS

In principle, our proposal is not in conflict with previous causal fairness approaches, and the two complement each other. Both our proposal and previous causal fairness approaches aim to model the data generating process, and both emphasize the procedural implications.

However, our proposal extends the scope of consideration beyond sensitive variables, and explicitly incorporates the influence of contextual environments. For instance, when operationalizing our proposal, we do not drop relevant variables, e.g., the `Address` of an individual, which is often omitted in previous literature (Kilbertus et al., 2017; Kusner et al., 2017; Nabi & Shpitser, 2018; Chiappa, 2019; Wu et al., 2019; Mary et al., 2019; Ding et al., 2021). Furthermore, the findings of our analyses suggest that we should utilize all information available, and furthermore, actively look for and develop better measurements for social determinants, so that we can better understand and address structural injustice. Future works naturally include the development of causal effect estimands that incorporate both sensitive attributes and social determinants, and our proposal and previous causal fairness approaches can be used in conjunction to achieve the goal.

### A.3. Additional Remarks on Related Terms

### A.3.1. THE USES OF "STRUCTURE" IN RELATED DISCIPLINES

The term "structure" and "structural" are utilized in different ways by related disciplines. For the literature of causal learning and reasoning, the term "structure" and "structural" are often used to describe how causal structures look like among variables of interest (Spirtes et al., 1993; Pearl, 2000; Peters et al., 2017; Hernán & Robins, 2020), e.g., in terms of causal graphs and/or structural equation models (SEMs). For the literature of structural justice and social determinants, the term "structural" is used to denote the systemic ways in which society is organized, e.g., through policies, laws, and social norms, that perpetuate discrimination and animus towards certain groups (Carmichael et al., 1967; Sowell, 1972; Tilly, 1998; Yearby, 2018; Robinson et al., 2020; Alexander, 2020; Yearby et al., 2022). There are interests in the social determinants of health literature to use DAGs as a tool for illustrative purposes, abstracting key concepts or areas that are interrelated at a high level, and modeling the mechanism through which structural forms of discriminations get realized (racism, sexism, etc.) (Robinson et al., 2020; Yearby et al., 2022).

### A.3.2. CONCEPTUAL DISTINCTIONS BETWEEN "UNFAIRNESS" AND "DISCRIMINATION"

The term "discrimination" refers to actions, practices, or policies that are based on the (perceived) social group membership of those affected. Standard accounts mandate that these groups are socially salient, i.e., they must significantly shape interactions within important social contexts (Lippert-Rasmussen, 2006; Porat, 2005; Al Ramiah et al., 2010) while recent works have challenged the social salience requirement (Eidelson, 2015). The term unfairness is typically understood as the broader concept, which encompassing any violation of principles of justice or proper treatment (Alexander, 1992; Moreau, 2020). In algorithmic fairness literature, existing fairness inquiries (including observational and causal ones) tend to gravitate towards quantifying discrimination. Meanwhile, *achieving* fairness as structural justice (through addressing *social determinants*) receives less attention compared to *enforcing* fairness as non-discrimination (through addressing *sensitive attributes*).

# B. Further Discussions on Social Determinants

In this section, we provide a variable audit protocol to operationalize Definition 2.2, and discuss the common presence of social determinants in various practical scenarios, where influence of contextual environments on individuals is often substantial.

## B.1. Operationalizing Definition of Social Determinants

**Variable Audit Protocol**  To operationalize Definition 2.2, we provide the following minimum variable audit protocol:

1. (Validate context-level definition) Confirm that each $S$ variable is defined at a context level (e.g., neighborhood, institution, jurisdiction), not as an individual attribute.
2. (Validate structural grounding of $S$) Provide a concrete mechanism linking variation in $S$ to social-structural processes (e.g., policy rules, institutional practices, resource allocation).
3. (Check exogenous stratification) Ensure that any aggregation used to compute $S$ relies on externally defined groupings (e.g., geographic or institutional boundaries), not clusters derived from individual features.
4. (Set and audit granularity) Re-evaluate results under coarser and finer versions of the same context definition (e.g., following the spirit of geographic aggregation sensitivity and de-identification analyses in census disclosure review).
5. (Slice reporting by $S$) Report key metrics (e.g., performance, empirical violations of standard sensitive-attribute-based fairness) stratified by $S$-defined contexts.
6. (Proxy correlation audit and mitigation) Quantify correlations between $S$ and sensitive attributes at the context level. If correlations exceed a threshold, either (i) coarsen $S$, (ii) remove or replace the variable, or (iii) explicitly justify its use and qualify downstream interpretations.
7. (Robustness to specification) Recompute $S$ under alternative plausible context definitions (e.g., different boundaries or institutional groupings). Verify that substantive conclusions remain stable across these specifications.
8. (Feedback loop and deployment monitoring) Track whether decisions informed by the model induce systematic changes in the distribution of $S$ across contexts.

**Caution Against Potential Misuses**  We would like to caution against potential misuses of social determinants:

1. (Re-identification via context linkage) Cross-referencing granular context units with external geospatial or administrative records to identify individuals, particularly when context membership is sparse.
2. (Proxy laundering at individual-/context- level) Deliberately adopting a social-determinant related variable as a legally permissible proxy for protected attributes (individual-level) or neighborhood (context-level), exploiting the context-level framing to enact discrimination while claiming neutrality.
3. (Strategic manipulation of context) Leveraging influence over context boundaries or policy (e.g., redistricting, institutional reclassification) to alter the value of social determinants and shift model outputs.

**Safeguard Against Proxy Laundering**  In practice, we can distinguish legitimate social determinants $S$ from proxy laundering through complementary safeguards:

1. Tests / diagnostics
   - (Proxy correlation audit) Measure correlation between $S$ and sensitive attributes at the context level. Treat high correlation as a risk signal.
   - (Boundary robustness) Recompute $S$ under alternative context definitions. Instability suggests incidental correlations rather than structural grounding.
2. Governance constraints
   - (Exogeneity requirement) Restrict $S$ to variables defined over externally specified contexts, and exclude groupings derived from individual features.
   - (Usage constraint) Limit $S$ to context-level auditing and prohibit individual-level decision-making use (unless a clear, documented justification is provided).
3. Documentation requirements
   - For each $S$, document: (i) construction and data sources, (ii) structural mechanism linking $S$ to social-structural processes, (iii) observed correlations with sensitive attributes, and (iv) any mitigation steps taken.

### B.2. Common Presence of Social Determinants

To strike a balance between a broad discussion and a case study, we considered a concrete empirical setting of college admissions in the main paper, and demonstrate the nuanced analyses our quantitative proposal facilitates. However, the implications of explicitly and carefully considering social determinants are not limited to the college admissions setting.

**Social Determinants – Health**   In terms of the influence of environments on individual's health, previous literature has considered how environmental hazards disproportionately affect low-income populations and communities of color (Warren et al., 2002), how indoor air pollution affects women and children in low-income regions (Manisalidis et al., 2020), and the structural implications of social determinants on how society should be organized (Robinson et al., 2020; Yearby et al., 2022). The time consistency and (sub-)population performance disparity of a mortality predictor has been evaluated in the context of advance care planning (Handler et al., 2023). More broadly, a review on economic research has also been conducted to show how environmental changes impact public health in both developed and developing countries (Remoundou & Koundouri, 2009).

**Social Determinants – Education**   In terms of the influence of environments on individual's educational attainments, previous literature has considered how the quality of schools and the availability of educational resources affect students' academic performance (Coleman, 1968; 1988), how the family and neighborhood environments influence education (Jencks, 1972), and implications of various affirmative-action policies (usually under different names) across countries with different histories and cultures (Sowell, 2004).

**Social Determinants – Employment**   In terms of the influence of environments on individual's employment opportunities, previous literature has considered the relation between the employment of residents and the rationalization and optimization level of region's industrial structures (Cao et al., 2017; Qin et al., 2022), the psychological perspective of (e.g., influence from collective values of community) job search behaviors (van Hooft et al., 2021), and how the employment rate of residents is influenced by job quality (Howell & Kalleberg, 2019).

## C. Background Information of Admission Strategies & Proofs of Theoretical Results

In this section, we provide background information of admission strategies, present additional theoretical results and the proofs.

### C.1. Background Information of Admission Strategies

**Quota-Based Admissions**   The quota-based admission is a type of affirmative-action admission strategy that sets specific limits on the number of admissions for applicants from different demographic backgrounds. This admission strategy was originally designed to rectify historical injustice by directly setting aside admission quotas to increase the representation of URM students. However, due to the rigid nature of the quota-based mechanism, this admission strategy has been controversial and addressed by the U.S. Supreme Court in the landmark case *University of California Regents v. Bakke (1978)* (Supreme Court, 1978). It was held that the use of strict racial quotas in college admission was unconstitutional, and was reaffirmed in another landmark case *Grutter v. Bollinger (2003)* (Supreme Court, 2003b).

**Holistic Review with Plus Factors**   Holistic review with plus factors is another type of affirmative-action admission strategy, involving consideration of multiple factors that together define each individual applicant. The key element of this process is the use of plus factors, where certain characteristics, for instance, race and ethnic group, are given additional weight to promote diversity in the student body and rectify historical disadvantages. This approach was upheld by the U.S. Supreme Court in *Grutter v. Bollinger (2003)* (Supreme Court, 2003b), but was overruled in recent decisions for *Students for Fair Admissions (SFFA) v. Harvard & UNC (2023)* (Supreme Court, 2023a;b), effectively banning race-conscious admissions.

For holistic review with plus factors, we model its affirmative-action emphasis on the URM group through a distribution shift, i.e., from the original scale to the plus-factor scale, instead of an automatic awarding of points for each URM applicant. Our modeling choice is for the purpose of avoiding the introduction of rigid and mechanical characteristics to the process, as was addressed in *Gratz v. Bollinger (2003)* (Supreme Court, 2003a).

**Top-Percentage Plans** The top-percentage plans are college admission policies that guarantee admission to students who graduate in a certain top percentage of their high school classes. The top-percentage plans are generally not considered traditional affirmative-action admission strategies. Instead, these policies are race-neutral alternatives aiming to promote diversity by drawing students from a wide range of schools with different socio-economic and geographic backgrounds, without explicitly considering race. A prominent example is the University of Texas's Top 10% Rule, which guarantees admission to students in the top 10% of their class. Another is the Eligibility in the Local Context (ELC) program of University of California, which was introduced after the 1996 California Proposition 209 banned the use of race, ethnicity, and gender in public university admissions in California.

## C.2. Proof of Theorem 4.5 in Section 4.2

**Theorem.** *Under Assumptions 4.1–4.4, let us denote with $\eta_{\text{quota}} \in \left[1, \frac{n}{n_a^{(\text{poor})} + n_a^{(\text{rich})}}\right]$ the weighting coefficient over the natural proportion of URM applicants in population, such that the quota for URM admissions in the selective college is $\eta_{\text{quota}} \cdot \left(\frac{n_a^{(\text{poor})} + n_a^{(\text{rich})}}{n} g\right)$. Then, the quota-based admission strategy imposes a more competitive requirements (in terms of score threshold) for non-URM applicants from the poor region, than that for URM applicants from the rich region, unless the following condition on region-specific academic preparedness CDF's is satisfied:*

$$\max_{q \in [0, \infty)} \frac{F^{(\text{rich})}(q)}{F^{(\text{poor})}(q)} \geq \frac{(n_{a'}^{(\text{poor})} + n_{a'}^{(\text{rich})}) \eta_{\text{quota}}}{(n_a^{(\text{poor})} + n_a^{(\text{rich})})(1 - \eta_{\text{quota}}) + (n_{a'}^{(\text{poor})} + n_{a'}^{(\text{rich})})} . \tag{C.1}$$

*Proof.* Quota-based admission reserves certain number of selective admission spots for the URM group, weighted by a coefficient $\eta_{\text{quota}} > 1$ over natural proportion of URM applicants, i.e., $\eta_{\text{quota}} \cdot \left(\frac{n_a^{(\text{poor})} + n_a^{(\text{rich})}}{n} g\right)$. Then, the available selective admission spots for the non-URM group is $g - \eta_{\text{quota}} \cdot \left(\frac{n_a^{(\text{poor})} + n_a^{(\text{rich})}}{n} g\right)$.

For the convenience of notation, let us denote $\eta'_{\text{quota}}$ the weight coefficients for the non-URM group over the natural proportion of non-URM applicants in the population, such that:

$$\eta'_{\text{quota}} \cdot \left(\frac{n_{a'}^{(\text{poor})} + n_{a'}^{(\text{rich})}}{n} g\right) = g - \eta_{\text{quota}} \cdot \left(\frac{n_a^{(\text{poor})} + n_a^{(\text{rich})}}{n} g\right), \tag{C.2}$$

Notice that $\eta'_{\text{quota}} \in [0, 1]$ since $\eta_{\text{quota}} \in \left[1, \frac{n}{n_a^{(\text{poor})} + n_a^{(\text{rich})}}\right]$. Additionally, $\eta'_{\text{quota}}$ is not an additional parameter whose value can vary freely, and it is fully determined by the numeric relation specified in Equation (C.2).

Because of the limited availability of selective admissions $g$, when employing the quota-based admission strategy, the score thresholds for each group will change as a result of the introduced quota requirements specified by weighting factors $\eta_{\text{quota}}$ and $\eta'_{\text{quota}}$. In particular, under Assumptions 4.1–4.4, the number of selective admissions for each group is calculated by the weighted sum (according to the probability of getting admitted to the selective college) of applicants from the group across regions, and the selective admission counts need to satisfy the quota requirements:

$$\begin{aligned} n_a^{(\text{poor})} \cdot F^{(\text{poor})}\left(q_a^{(poor)}\right) + n_a^{(\text{rich})} \cdot F^{(\text{rich})}\left(q_a^{(rich)}\right) &= \eta_{\text{quota}} \cdot \left(\frac{n_a^{(\text{poor})} + n_a^{(\text{rich})}}{n} g\right), \\ n_{a'}^{(\text{poor})} \cdot F^{(\text{poor})}\left(q_{a'}^{(poor)}\right) + n_{a'}^{(\text{rich})} \cdot F^{(\text{rich})}\left(q_{a'}^{(rich)}\right) &= \eta'_{\text{quota}} \cdot \left(\frac{n_{a'}^{(\text{poor})} + n_{a'}^{(\text{rich})}}{n} g\right). \end{aligned} \tag{C.3}$$

Since the quota-based admission strategy ensures Equation (C.3) is satisfied given the region-specific demographic makeup (Assumption 4.1), we have:

$$F^{(\text{poor})}\left(q_a^{(poor)}\right) = \frac{g \cdot \eta_{\text{quota}}}{n} = F^{(\text{rich})}\left(q_a^{(rich)}\right), \tag{C.4}$$

$$F^{(\text{poor})}\left(q_{a'}^{(poor)}\right) = \frac{g \cdot \eta'_{\text{quota}}}{n} = F^{(\text{rich})}\left(q_{a'}^{(rich)}\right). \tag{C.5}$$

Let us consider the left-hand-side (LHS) and right-hand-side (RHS) of each equation.

- LHS equals to RHS of Equation (C.4): since $F^{(\text{rich})}$ dominates $F^{(\text{poor})}$ (Assumption 4.3), we have $q_a^{(poor)} > q_a^{(rich)}$, i.e., among URM applicants, the threshold for the raw score in the poor region is lower than that for the rich region.

- LHS equals to RHS of Equation (C.5): for the same reason as above, we have $q_{a'}^{(poor)} > q_{a'}^{(rich)}$, i.e., among non-URM applicants, the threshold for the raw score in the poor region is lower than that for the rich region.
- LHS of Equation (C.4) and LHS of Equation (C.5): since $\eta'_{\text{quota}} < 1 < \eta_{\text{quota}}$, we have $q_a^{(poor)} > q_{a'}^{(poor)}$, i.e., for the poor region, the threshold for the raw score of URM applicants is lower than that for non-URM applicants.
- RHS of Equation (C.4) and RHS of Equation (C.5): for the same reason as above, we have $q_a^{(rich)} > q_{a'}^{(rich)}$, i.e., for the rich region, the threshold for the raw score of URM applicants is lower than that for non-URM applicants.

However, the relative magnitude relation between $q_{a'}^{(poor)}$ (for non-URM applicants residing in the poor region) and $q_a^{(rich)}$ (for URM applicants residing in the rich region) can go either way. Specifically, we can show that if $\max_{q \in [0,\infty)} \frac{F^{(\text{rich})}(q)}{F^{(\text{poor})}(q)} < \frac{\eta_{\text{quota}}}{\eta'_{\text{quota}}}$, then $q_{a'}^{(poor)} < q_a^{(rich)}$, i.e., the threshold at the raw score for non-URM applicants in the poor region is higher than that for URM applicants from the rich region:

$$\text{when} \max_{q \in [0,\infty)} \frac{F^{(\text{rich})}(q)}{F^{(\text{poor})}(q)} < \frac{\eta_{\text{quota}}}{\eta'_{\text{quota}}}, \text{ we have } \frac{\eta_{\text{quota}}}{\eta'_{\text{quota}}} \cdot F^{(\text{poor})}(q_{a'}^{(poor)}) > F^{(\text{rich})}(q_{a'}^{(poor)}), \tag{C.6}$$

and at the same time

$$\frac{\eta_{\text{quota}}}{\eta'_{\text{quota}}} \cdot F^{(\text{poor})}(q_{a'}^{(poor)}) \overset{(i)}{=} F^{(\text{poor})}(q_a^{(poor)}) \overset{(ii)}{=} F^{(\text{rich})}(q_a^{(rich)}), \tag{C.7}$$

where (i) results from Equation (C.4) and Equation (C.5), and (ii) follows Equation (C.4).

Because $F^{(\text{rich})}(q_a^{(rich)}) > F^{(\text{rich})}(q_{a'}^{(poor)})$ and the CDF function $F^{(\text{rich})}(\cdot)$ is non-decreasing, we have $q_{a'}^{(poor)} < q_a^{(rich)}$. In other words, as a necessary condition to prevent this, we need

$$\max_{q \in [0,\infty)} \frac{F^{(\text{rich})}(q)}{F^{(\text{poor})}(q)} \geq \frac{\eta_{\text{quota}}}{\eta'_{\text{quota}}}, \tag{C.8}$$

after re-arranging, and incorporating Equation (C.2), gives us

$$\max_{q \in [0,\infty)} \frac{F^{(\text{rich})}(q)}{F^{(\text{poor})}(q)} \geq \frac{(n_{a'}^{(\text{poor})} + n_{a'}^{(\text{rich})})\eta_{\text{quota}}}{(n_a^{(\text{poor})} + n_a^{(\text{rich})})(1 - \eta_{\text{quota}}) + (n_{a'}^{(\text{poor})} + n_{a'}^{(\text{rich})})} \,.$$

$\square$

## C.3. Additional Theoretical Result Related to Holistic Review with Plus Factors

For the purpose of deriving a closed-from characterization, we introduce an additional quantitative assumption:

**Assumption C.1** (Gamma Parameterization of Academic Preparedness Distribution). Let $S$ denote the non-negative overall academic index score of an applicant's academic preparedness. Further let $S_{\text{MAX}}$ and $S_{\text{MIN}}$ denote the highest and lowest possible values of the score. Within any specific region $r \in \{\text{poor}, \text{rich}\}$, the log-converted relative score $Q$ is Gamma distributed with region-specific shape and scale parameters, $k^{(r)}$ and $\theta^{(r)}$, respectively. Furthermore, the rich region's cumulative distribution function (CDF) of log-converted relative score $Q$ dominates that of the poor region:

$$Q \sim \Gamma\big(k^{(r)}, \theta^{(r)}\big), \text{ where } Q := -\log\left(\frac{S - S_{\text{MIN}}}{S_{\text{MAX}} - S_{\text{MIN}}}\right),$$
$$\forall q \in [0, \infty), \ F^{(\text{rich})}(q) \geq F^{(\text{poor})}(q),$$
$$\text{where } F^{(r)}(q) \text{ is the CDF of } \Gamma\big(k^{(r)}, \theta^{(r)}\big), r \in \{\text{poor}, \text{rich}\}.$$

In Assumption C.1, the conversion of the score maps the domain of values $[S_{\text{MIN}}, S_{\text{MAX}}]$ (higher score $S$ is more competitive) to $[0, \infty)$, where the closer to $0$ the converted score $Q$, the more competitive. We selected Gamma distributions for their flexibility in capturing the one-sided skew commonly observed in academic performance distributions, consistent with prior educational assessment research (Campos, 2013; Sari et al., 2025).

Putting aside the evolving jurisprudence (Supreme Court, 2003b; 2023a;b), we aim to precisely characterize holistic review in terms of its implications on the distribution of benefits and burdens among individuals, when allocating the limited spots in selective college admissions. When taking into account of social determinants signified by Region, we show that holistic review with plus factors may benefit applicants from better-off areas more than those from less well-off areas:

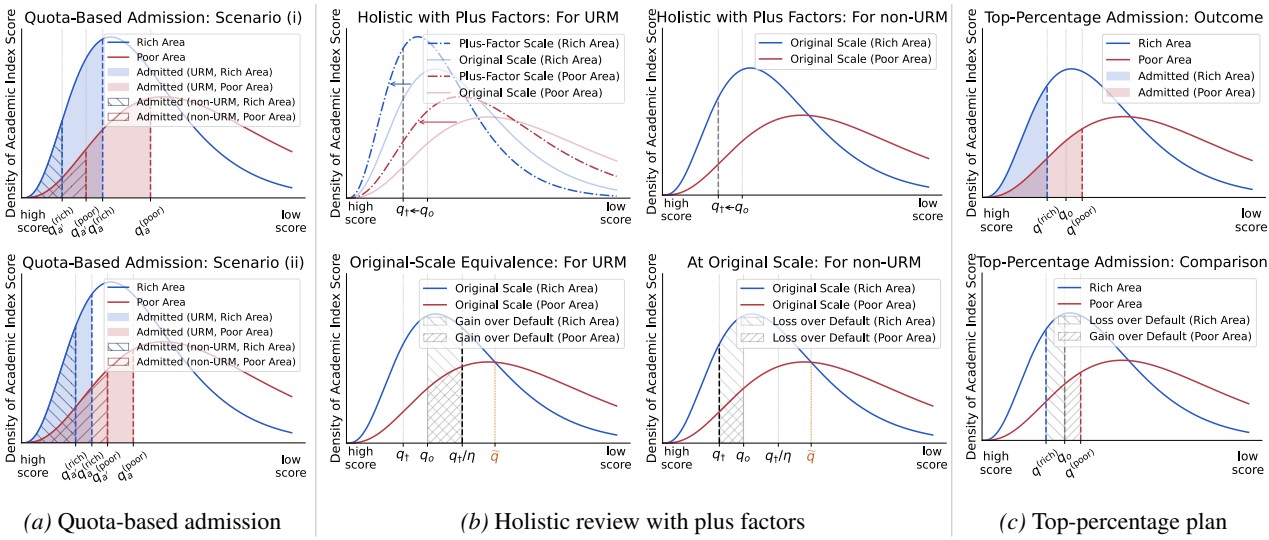

*Figure 3.* Fairness implications of different admission strategies. Panel (a): quota-based admission can introduce additional unfairness against non-URM applicants from the poor region. Panel (b): holistic review with plus factors tends to benefit URM applicants in the rich region more than these in the poor region. Panel (c): top-percentage plan transfer admission opportunity from the rich region to the poor region, and the redistribution is proportional to the natural region-specific demographic compositions.

**Theorem C.2** (Holistic Review with Plus Factors Benefits URM in Rich Region More). *Under Assumptions 4.1–4.4 and also C.1, let us denote with $\eta_\dagger < 1$ the multiplicative coefficient on the scale parameter of Gamma distributions for URM applicants' academic index scores, such that the perceived scores of URM applicants shift more probability density towards the high-score end. Let us denote with $q_o$ the default threshold for selective admission, and with $q_\dagger$ the threshold if the admission procedure is a holistic review with plus factors. Further assume that region-specific shape parameters satisfy $k^{(\mathrm{poor})} = k^{(\mathrm{rich})} = k_o$. Then, the increase in the probability of selective admission for URM applicants from the rich region, is larger than that for URM applicants from the poor region:*

$$\text{if the selective admission is limited in availability such that } q_o < \frac{k_o \ln(\theta^{(\mathrm{poor})}/\theta^{(\mathrm{rich})})}{1/\theta^{(\mathrm{rich})} - 1/\theta^{(\mathrm{poor})}},$$

$$\text{then } \forall \eta_\dagger \geq \frac{q_o(1/\theta^{(\mathrm{rich})} - 1/\theta^{(\mathrm{poor})})}{k_o \ln(\theta^{(\mathrm{poor})}/\theta^{(\mathrm{rich})})}, F^{(\mathrm{rich})}(q_\dagger/\eta_\dagger) - F^{(\mathrm{rich})}(q_o) > F^{(\mathrm{poor})}(q_\dagger/\eta_\dagger) - F^{(\mathrm{poor})}(q_o).$$

*Proof.* The holistic review with plus factors changes the scale parameter of the Gamma distribution corresponding to URM applicants' academic index scores, from the original scale, i.e., $\Gamma(k_o, \theta^{(r)})$, to the plus-factor scale, i.e., $\Gamma(k_o, \eta_\dagger \cdot \theta^{(r)})$, where $r \in \{\mathrm{poor}, \mathrm{rich}\}$. The admission procedure does not change how non-URM applicants' scores are perceived, i.e., it remains at the original scale, $\Gamma(k_o, \theta^{(r)})$.

Then, we can calculate the default threshold $q_o$ and that when the admission strategy is employed, $q_\dagger$, as follows:

$$(n_a^{(\mathrm{poor})} + n_{a'}^{(\mathrm{poor})}) \cdot F^{(\mathrm{poor})}(q_o) + (n_a^{(\mathrm{rich})} + n_{a'}^{(\mathrm{rich})}) \cdot F^{(\mathrm{rich})}(q_o) = g, \tag{C.9}$$

$$n_a^{(\mathrm{poor})} \cdot F_\dagger^{(\mathrm{poor})}(q_\dagger) + n_a^{(\mathrm{rich})} \cdot F_\dagger^{(\mathrm{rich})}(q_\dagger) + n_{a'}^{(\mathrm{poor})} \cdot F^{(\mathrm{poor})}(q_\dagger) + n_{a'}^{(\mathrm{rich})} \cdot F^{(\mathrm{rich})}(q_\dagger) = g, \tag{C.10}$$

where $F^{(r)}(\cdot)$ is the CDF of $\Gamma(k_o, \theta^{(r)})$, and $F_\dagger^{(r)}(\cdot)$ is that of $\Gamma(k_o, \eta_\dagger \cdot \theta^{(r)})$.

Because of the numerical property of Gamma CDF's, we have:

$$\forall q \in [0, \infty), \quad F_\dagger^{(r)}(q) = \frac{1}{\Gamma(k)} \gamma\left(k_o, \frac{q}{\eta_\dagger \cdot \theta^{(r)}}\right) = \frac{1}{\Gamma(k)} \gamma\left(k_o, \frac{q/\eta_\dagger}{\theta^{(r)}}\right) = F^{(r)}\left(\frac{q}{\eta_\dagger}\right), \tag{C.11}$$

where $\gamma(\cdot, \cdot)$ is the incomplete gamma function. In other words, when employing holistic review with plus factors, having the same threshold $q_\dagger$ operating on $F_\dagger^{(r)}(\cdot)$ for URM applicants and $F^{(r)}(\cdot)$ for non-URM applicants, is equivalent to having a threshold $q_\dagger/\eta_\dagger$ for URM applicants and $q_\dagger$ for non-URM applicants but operating only on $F^{(r)}(\cdot)$, where $q_\dagger/\eta_\dagger > q_o > q_\dagger$.

Since $k^{(\text{poor})} = k^{(\text{rich})} = k_o$, the two PDF curves only have one intersecting point:

$$\frac{1}{\Gamma(k_o)(\theta^{(\text{poor})})^{k_o}}q^{k_o-1}e^{-q/\theta^{(\text{poor})}} = \frac{1}{\Gamma(k_o)(\theta^{(\text{rich})})^{k_o}}q^{k_o-1}e^{-q/\theta^{(\text{rich})}}$$

$$\implies \quad q = \frac{k_o \ln(\theta^{(\text{poor})}/\theta^{(\text{rich})})}{1/\theta^{(\text{rich})} - 1/\theta^{(\text{poor})}}. \tag{C.12}$$

Then, when the selective admission availability is limited such that $q_o < \frac{k_o \ln(\theta^{(\text{poor})}/\theta^{(\text{rich})})}{1/\theta^{(\text{rich})} - 1/\theta^{(\text{poor})}}$, because of the CDF dominance of the rich region over the poor region (Assumption C.1), and that we can equivalently compare thresholds $q_\dagger/\eta_\dagger > q_o > q_\dagger$ at the original-scale CDF $F^{(r)}(\cdot)$, we have:

$$\forall \eta_\dagger \in \left[ \frac{q_o(1/\theta^{(\text{rich})} - 1/\theta^{(\text{poor})})}{k_o \ln(\theta^{(\text{poor})}/\theta^{(\text{rich})})}, 1 \right), F^{(\text{rich})}\left(\frac{q_\dagger}{\eta_\dagger}\right) - F^{(\text{rich})}(q_o) > F^{(\text{poor})}\left(\frac{q_\dagger}{\eta_\dagger}\right) - F^{(\text{poor})}(q_o).$$

$$\square$$

Theorem C.2 characterizes different levels of benefits for URM applicants from different regions. Specifically, in terms of the increase in admission probability to the selective college, URM applicants from the rich region benefit more from the admission procedure that utilizes holistic review with plus factors, compared to URM applicants from the poor region. To better demonstrate our theoretical result, we provide illustrations in Figure 3(b).

As presented in top-row subfigures in Figure 3(b), at the original scale, the region-specific distributions of academic preparedness are the same for URM and non-URM applicants (Assumption 4.2). Holistic review with plus factors grants preference to URM applicants by perceiving their scores, at the distribution level, as if they were sampled from a distribution that is more concentrated at the high-score end (the plus-factor scale). Because of the limited availability in selective admissions, the threshold $q_\dagger$ for admission under holistic review with plus factors is more competitive than the default $q_o$, i.e., $q_\dagger < q_o$, for both URM and non-URM applicants. While non-URM applicants are assessed on the original scale, URM applicants are evaluated on a plus-factor scale. Under the Gamma parameterization (Assumption C.1), this is equivalent to employing a more competitive threshold $q_\dagger$ for non-URM applicants but a less competitive one $q_\dagger/\eta_\dagger$ for URM applicants, where $q_\dagger < q_o < q_\dagger/\eta_\dagger$. Although the mathematical form of $q_o < {k_o \ln(\theta^{(\text{poor})}/\theta^{(\text{rich})})}/{(1/\theta^{(\text{rich})} - 1/\theta^{(\text{poor})})}$ appears convoluted, the condition itself is relatively mild. Graphically speaking, the spots at the selective college are limited such that the threshold $q_o$ does not reach the point where region-specific Gamma density curves (in the original scale) intersect, as depicted by $\widetilde{q}$ in Figure 3(b).

From the shaded areas in bottom-row subfigures in Figure 3(b), we can see that the increased admission probability for URM groups comes with a corresponding reduction in that for non-URM groups. However, such redistribution benefits URM applicants in the rich region more than those in the poor region, essentially disadvantaging URM applicants in less socio-economically advantaged areas.

## C.4. Additional Theoretical Result Related to the Top-Percentage Plans

Taking into account the demographic composition of applicants and the number of available spots at the selective college, we characterize the difference between top-percentage plans compared to the default selective admission. When explicitly considering the role of `Region` in applicants' academic preparedness, we show that the redistribution of limited selective admissions, as implied by top-percentage plans, is carried out by reallocating availability from the rich region to the poor region, regardless of the demographic group of applicants:

**Theorem C.3** (Top-Percentage Plans Reallocate Spots from Rich Region to Poor Region). *Under Assumptions 4.1–4.4 and also C.1, let us denote with $q_o$ the default threshold for selective admission, and with $q^{(\text{poor})}$ and $q^{(\text{rich})}$ the thresholds for poor and rich regions, respectively, if top-percentage plans are employed. Then, the increase in selective admissions (in terms of counts) for applicants from the poor region, comes from spots reallocated out of the rich region. This redistribution is a result of the top-percentage plans, and is not relevant to applicants' demographic group:*

$$\left(n_a^{(\text{poor})} + n_{a'}^{(\text{poor})}\right)\left[F^{(\text{poor})}(q^{(\text{poor})}) - F^{(\text{poor})}(q^{(o)})\right] = \left(n_a^{(\text{rich})} + n_{a'}^{(\text{rich})}\right)\left[F^{(\text{rich})}(q^{(o)}) - F^{(\text{rich})}(q^{(\text{rich})})\right].$$

*Furthermore, if region-specific shape parameters satisfy $k^{(\text{poor})} = k^{(\text{rich})}$, we additionally have:*

$$q^{(\text{poor})}/q^{(\text{rich})} = \theta^{(\text{poor})}/\theta^{(\text{rich})}.$$

*Proof.* Top-percentage plans distribute the limited availability of selective admissions in a way that guarantee admissions to top-percentage applicants in their regions, and the resulting thresholds are region-specific. Then, we can calculate the default threshold $q_o$ and the region-specific thresholds when top-percentage plans are employed:

$$(n_a^{(\text{poor})} + n_{a'}^{(\text{poor})}) \cdot F^{(\text{poor})}(q_o) + (n_a^{(\text{rich})} + n_{a'}^{(\text{rich})}) \cdot F^{(\text{rich})}(q_o) = g, \tag{C.13}$$

$$(n_a^{(\text{poor})} + n_{a'}^{(\text{poor})}) \cdot F^{(\text{poor})}(q^{(\text{poor})}) + (n_a^{(\text{rich})} + n_{a'}^{(\text{rich})}) \cdot F^{(\text{rich})}(q^{(\text{rich})}) = g,$$

$$\text{where} \quad F^{(\text{poor})}(q^{(\text{poor})}) = F^{(\text{rich})}(q^{(\text{rich})}) = \frac{g}{n_a^{(\text{poor})} + n_a^{(\text{rich})} + n_{a'}^{(\text{poor})} + n_{a'}^{(\text{rich})}}. \tag{C.14}$$

Compare Equation (C.13) and Equation (C.14), we have:

$$(n_a^{(\text{poor})} + n_{a'}^{(\text{poor})}) \big[ F^{(\text{poor})}(q^{(\text{poor})}) - F^{(\text{poor})}(q^{(o)}) \big] = (n_a^{(\text{rich})} + n_{a'}^{(\text{rich})}) \big[ F^{(\text{rich})}(q^{(o)}) - F^{(\text{rich})}(q^{(\text{rich})}) \big].$$

Because of the numerical property of Gamma CDF's (as we have seen in the proof for Theorem C.2), when region-specific shape parameters satisfy $k^{(\text{poor})} = k^{(\text{rich})} = k$, we have:

$$F^{(\text{poor})}(q^{(\text{poor})}) = \frac{1}{\Gamma(k)} \gamma\big(k, \frac{q^{(\text{poor})}}{\theta^{(\text{poor})}}\big),$$

$$F^{(\text{rich})}(q^{(\text{rich})}) = \frac{1}{\Gamma(k)} \gamma\big(k, \frac{q^{(\text{rich})}}{\theta^{(\text{rich})}}\big),$$

together with Equation (C.14), and we have:

$$F^{(\text{poor})}(q^{(\text{poor})}) = F^{(\text{rich})}(q^{(\text{rich})}) \implies \frac{q^{(\text{poor})}}{\theta^{(\text{poor})}} = \frac{q^{(\text{rich})}}{\theta^{(\text{rich})}}, \text{ i.e., } \frac{q^{(\text{poor})}}{q^{(\text{rich})}} = \frac{\theta^{(\text{poor})}}{\theta^{(\text{rich})}}.$$

$\square$

Theorem C.3 characterizes the reallocation of the selective admission spots performed by top-percentage plans. In Figure 3(c), we use shaded areas to illustrate the transfer of admission opportunity (in terms of the region-wise probability of selective admission) from the rich region to the poor region. The additional selective admissions gained by the poor region, compared to default setting, are distributed proportionally to the natural demographic composition of each group.

## D. Additional Data Analyses on U.S. Census Data

In this section, we first demonstrate that enforcing group fairness metric while including social determinants (or their proxies) as features does not directly address structural injustice. Then, we present the age structure, racial composition, occupation distribution in different PUMAs.

### D.1. Sanity Check: Enforcing Demographic Parity with Social Determinants as Features

We consider the *Demographic Parity* (Calders et al., 2009; Dwork et al., 2012; Zemel et al., 2013; Feldman et al., 2015) notion of group fairness and a U.S. Census Data-based income prediction task (Ding et al., 2021). We consider a a $2 \times 2 \times 3$

*Table 2.* Effect of DP Enforcement on Accuracy and Racial Disparity

| Predictor | Accuracy (baseline) | Accuracy (DP enforced) | DP metric (baseline) | DP metric (DP enforced) |
|---|---|---|---|---|
| Logistic Regression (w/out ADI) | 0.785 | 0.742 | 0.130 | 0.012 |
| Logistic Regression (w/ ADI) | 0.784 | 0.736 | 0.130 | 0.017 |
| Random Forest (w/out ADI) | 0.806 | 0.804 | 0.115 | 0.025 |
| Random Forest (w/ ADI) | 0.806 | 0.807 | 0.120 | 0.014 |
| XGBoost (w/out ADI) | 0.814 | 0.810 | 0.140 | 0.025 |
| XGBoost (w/ ADI) | 0.817 | 0.814 | 0.148 | 0.025 |

*Table 3.* Racial Disparity Stratified by Social Determinant After DP Enforcement

| Predictor | ADI Region | African American | White | Asian | DP metric |
|---|---|---|---|---|---|
| Logistic Regression (w/out ADI) | Low | 0.697 | 0.693 | 0.690 | 0.008 |
| | Mid | 0.642 | 0.620 | 0.583 | 0.059 |
| | High | 0.545 | 0.566 | 0.455 | 0.111 |
| Logistic Regression (w/ ADI) | Low | 0.762 | 0.722 | 0.734 | 0.040 |
| | Mid | 0.652 | 0.648 | 0.594 | 0.058 |
| | High | 0.463 | 0.540 | 0.364 | 0.176 |
| Random Forest (w/out ADI) | Low | 0.651 | 0.713 | 0.693 | 0.062 |
| | Mid | 0.622 | 0.615 | 0.552 | 0.070 |
| | High | 0.545 | 0.549 | 0.455 | 0.094 |
| Random Forest (w/ ADI) | Low | 0.679 | 0.730 | 0.709 | 0.051 |
| | Mid | 0.630 | 0.602 | 0.552 | 0.077 |
| | High | 0.480 | 0.490 | 0.443 | 0.048 |
| XGBoost (w/out ADI) | Low | 0.725 | 0.698 | 0.681 | 0.044 |
| | Mid | 0.625 | 0.600 | 0.536 | 0.089 |
| | High | 0.504 | 0.517 | 0.419 | 0.098 |
| XGBoost (w/ ADI) | Low | 0.688 | 0.730 | 0.708 | 0.041 |
| | Mid | 0.627 | 0.594 | 0.512 | 0.116 |
| | High | 0.504 | 0.477 | 0.376 | 0.129 |

design: whether Demographic Parity (DP) with respect to race is enforced, and whether a social determinant (we use ADI) is included as a feature, across 3 model classes (logistic regression, random forest, and XGBoost).

From Tables 2 and 3, we can see that (1) including ADI as a feature does not consistently improve DP, which can sometimes worsen observed DP violations, and (2) when outcomes are stratified by social determinant, disparities become more pronounced.

Both findings are expected and support our position: standard sensitive-attribute-based fairness metrics effectively smooth over structural contexts, thereby obscuring variation across them. Moreover, including social determinants as predictive features, even when appropriate, is not equivalent to treating them as an evaluative axis for fairness.

### D.2. Age Structure of Population in PUMAs

In Figure 4, we present age distributions in different PUMAs. For instance, PUMAs 3729, 7503, 11300 show noticeable concentrations of younger individuals, particularly in the 20–40 age range, suggesting a potentially more dynamic, working-age population which may affect local labor markets and educational demands. In contrast, PUMAs 7318 and 11106 exhibit a more balanced distribution across age groups, but with a slight skew towards middle-aged populations, which could indicate stable, established communities possibly with higher home ownership and lower school enrollment rates. For PUMA 8512, there are peaks in the 20s and again in the 50s, represent a mix of young adults possibly associated with entry-level professional work, and also senior adults in established careers or nearing retirement. The age distribution for PUMA 300 shows a peak around the age of 70s, reflecting a demographic profile with a substantial proportion of senior adults. Each area's age distribution can profoundly impact local policies, economic conditions, and community services tailored to the dominant age groups' needs. Therefore, the residents will be positioned differently in terms of social determinants such as educational resources, employment opportunities, and healthcare providers.

### D.3. Racial Composition in PUMAs

In Figure 5, we present racial compositions across PUMAs. In the context of U.S. Census data, "Hispanic or Latino(a)" origin is considered an ethnicity, not a race. Individuals of Hispanic or Latino(a) origin can be of any race and are often asked to identify both their race and their ethnicity during the data collection. Therefore, the racial composition does not contain a separate category for Hispanic or Latino(a) individuals.

As we can see, for historical and cultural reasons, the racial compositions vary quite a bit across different regions. For instance, PUMA 5700 predominantly consists of White individuals, making up $84.1\%$ of its population, indicating a less

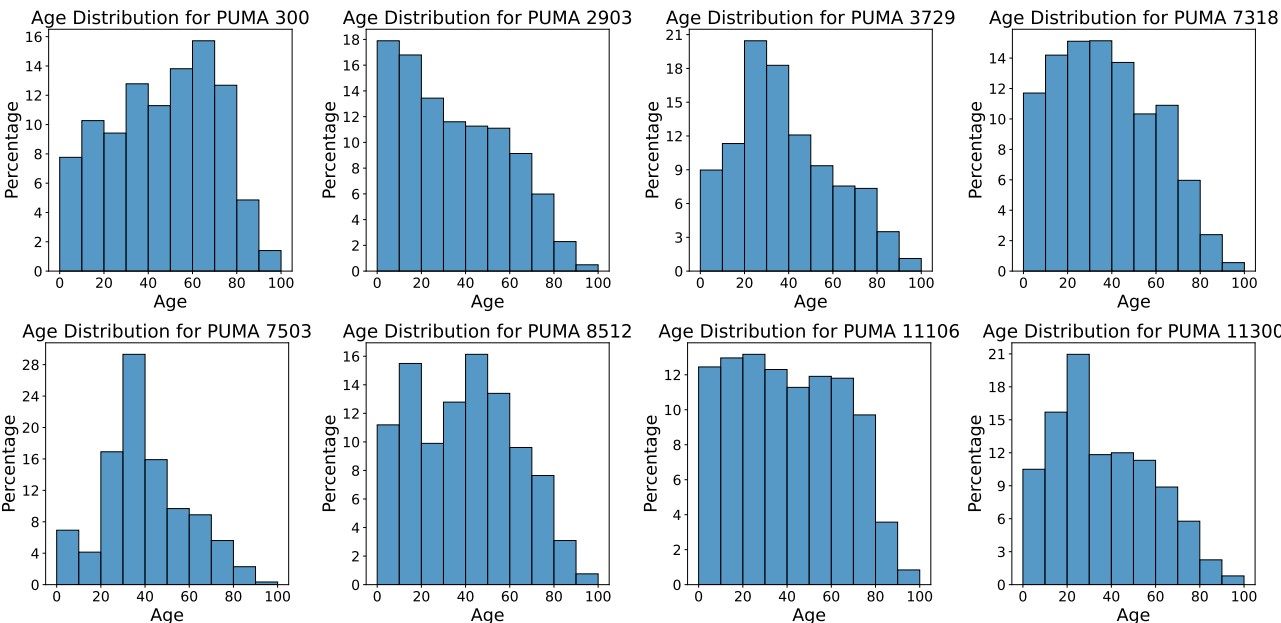

*Figure 4.* Age distribution in different PUMA regions in California based on US Census data.

racially diverse area compared to others. Similarly, PUMA 8504 displays a vast majority of Asian residents, accounting for 70.7% of the population. In contrast, PUMA 7318 offers a more balanced racial mix with no single group exceeding more than 30%, suggesting a more racially integrated community. These variations in racial composition can impact community needs, including educational services, cultural programs, and language services, and may influence local policy-making and resource allocation. Therefore, the association between social determinants and racial composition of the population can differ significantly across regions.

### D.4. Occupation Distribution in PUMAs

In Figure 6, we present distribution of occupations from certain categories in various PUMAs. The diverse workforce compositions reflect varying regional economic profiles and potential educational infrastructures. For instance, PUMAs 101 and 8503 display a strong presence of occupations related to science, engineering, education, and so on. In contrast, PUMA 6712 shows a more balanced distribution across different occupation categories (except for primary industries), suggesting a balanced mix of professional services and healthcare employment sectors. In terms of the category of farming, fishing, and forestry occupations, PUMAs 1901 and 8301 differ from other PUMAs (e.g., 101 and 8503). This category forms a significant part of the workforce (more than a third in both 1901 and 8301), reflecting an economy heavily reliant on primary industries. These patterns highlight how local natural and industrial resources, as well as economies, can significantly influence the occupational structures and, by extension, the training and education needed to support these sectors. Therefore, the social determinants in different regions can be shaped differently.

### D.5. Combination of Demographic Factors in PUMA

In Figure 7, we present how PUMAs can have very different profiles in terms of residents' age structure, racial decomposition, and occupation distribution. In terms of the age structure, PUMAs 3749 and 8504 show more concentrations in the 20–40 age range, while PUMA 1700 has a high proportion of senior adults. In terms of the race decomposition, the majority of residents are white (75.9%) for PUMA 1700, African American (41.5%), and Asian (70.7%) for PUMA 8504. In terms of the occupation distribution, while the proportion of medical and healthcare practitioners is similar across the three regions, the occupational structures are very different. For instance, nearly one half of the working force in PUMA 8504 is within the category of computer and mathematical occupations, while the number is significantly lower in PUMAs 1700 and 3749, with a proportion of 18.4% and 5.8%, respectively. Therefore, in addition to sensitive attributes, the comprehensive understanding of the social determinants in different regions can help inform policy-making and resource allocation decisions, so that we can achieve algorithmic fairness by taking into account both sensitive attributes and social determinants.

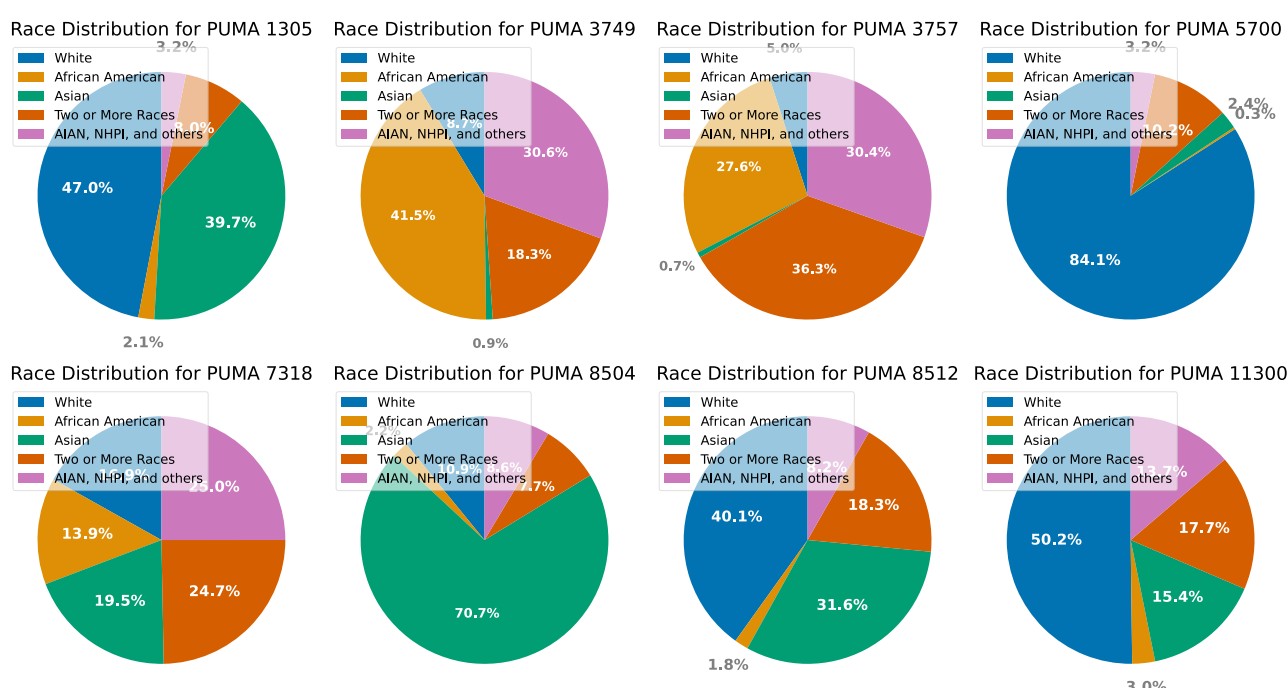

*Figure 5.* Racial composition in various PUMA regions in California based on U.S. Census data.

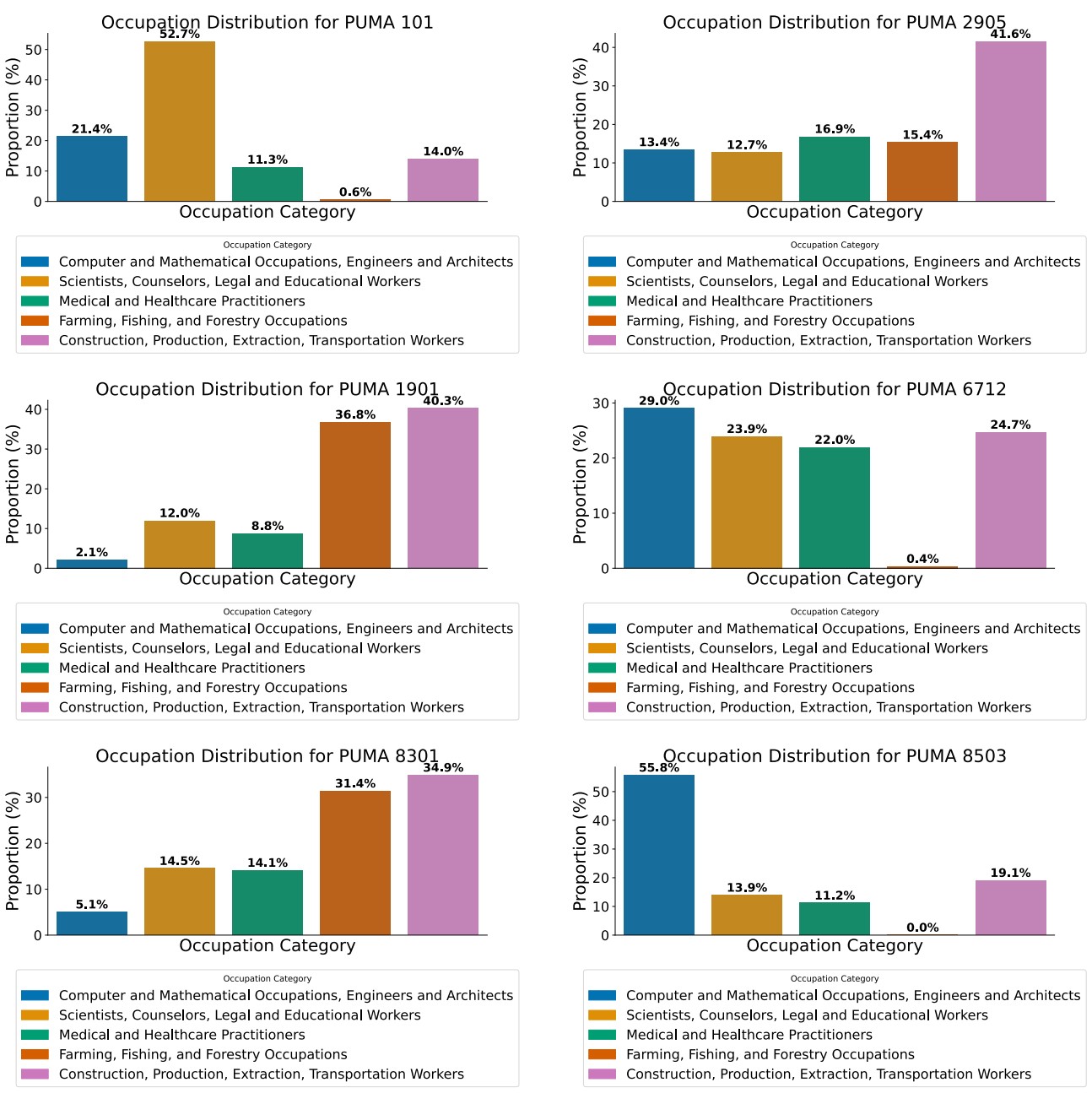

*Figure 6.* Occupational structure in various PUMA regions in California based on U.S. Census data.

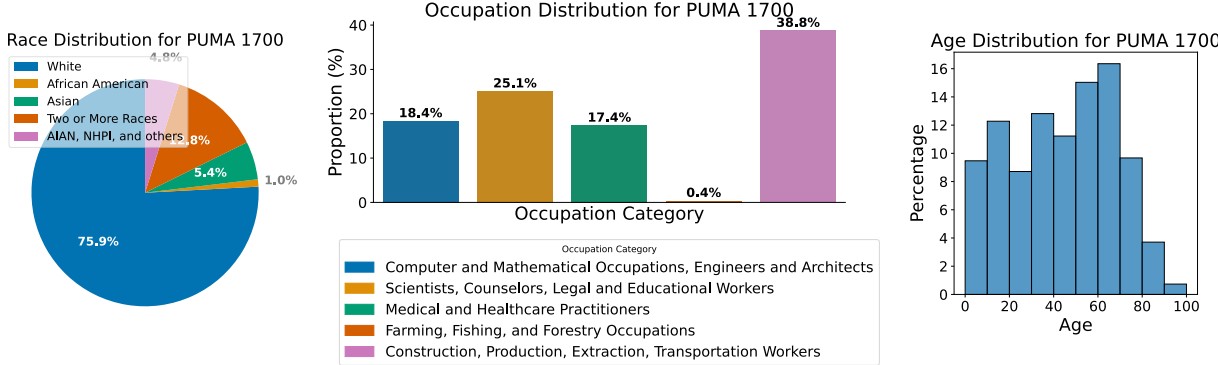

*(a)* PUMA 1700: racial decomposition, occupation distribution, and age structure.

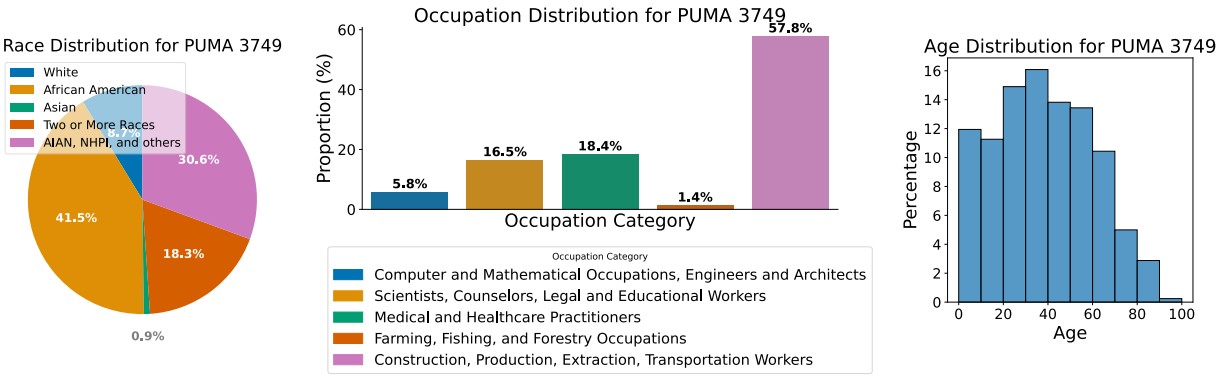

*(b)* PUMA 3749: racial decomposition, occupation distribution, and age structure.

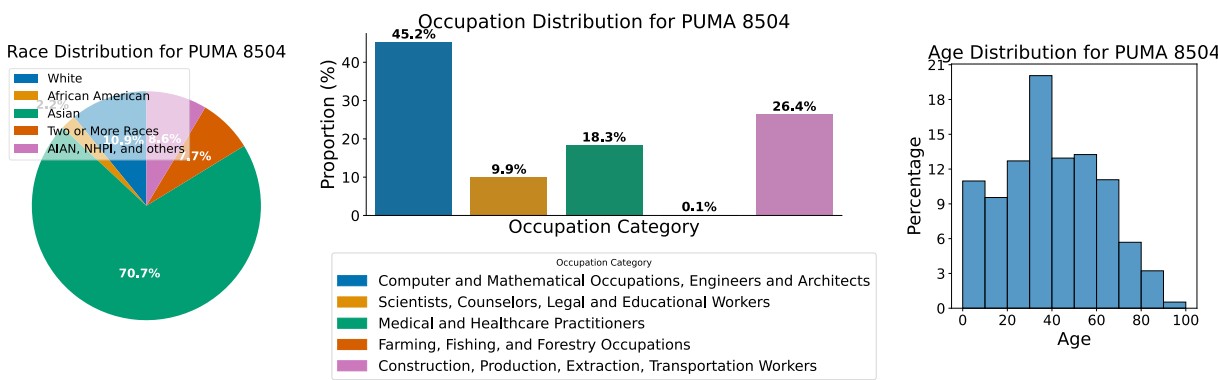

*(c)* PUMA 8504: racial decomposition, occupation distribution, and age structure.

*Figure 7.* PUMAs with different profiles in terms of residents' age, race, and occupation.

