# OpenReview forum: "Position: Beyond Sensitive Attributes, ML Fairness Should Quantify Structural Injustice via Social Determinants"
_ICML.cc/2026/Position_Paper_Track — ICML 2026 Position Paper Track regular_

### Official Review · Reviewer_6yWC · 2026-03-04

**Significance:** 3
**Argument Clarity:** 3
**Rating:** 5
**Confidence:** 3

**Questions:**

- Q1: Could you provide a more nuanced clarification of Definition 2.2, specifically regarding the complex interplay and the conceptual boundary between "Social Determinants" and "Sensitive Attributes" in highly entangled data generating processes? Illustrating this distinction with a concrete, real-world scenario would significantly enhance the intuitiveness and clarity of the proposed definition.

- Q2: Could the authors elaborate on how the proposed framework can be generalized to quantify non-spatial structural injustices, moving beyond the geographic proxies utilized in the current examples?

**Alternative Views Section:**

Yes

**Compliance With Llm Reviewing Policy A Conservative:**

Affirmed.

**Discussion Potential:**

3

**Final Justification:**

The authors’ rebuttal has addressed my concerns, and I support the acceptance of the current manuscript.

**Paper Summary:**

This position paper argues that machine learning fairness should evolve beyond its narrow focus on individual sensitive attributes (e.g., race, gender) to explicitly quantify and mitigate structural injustice through social determinants, such as community resources and environmental contexts.

**Position:**

Yes

**Position In Title:**

Yes

**Related Work:**

3

**Strengths And Weaknesses:**

Strengths:

- S1: The paper is well-written and easy to follow. The structure is logical, and the inclusion of clear "Takeaways" at the end of each section provides significant value in guiding the reader through complex conceptual shifts.

- S2: The empirical analyses are highly valuable and forcefully substantiate the paper's core arguments. By demonstrating the impact of social determinants in real-world, high-stakes domains, the paper bridges the gap between abstract theory and practical necessity.


Weaknesses:

- W1: The formal definition of Social Determinants (Definition 2.2) as simply a variable representing an aspect of the data generating process feels somewhat reductive. The manuscript would benefit from a deeper, more rigorous discussion of the complex relationship and boundaries between Social Determinants and Sensitive Attributes.

- W2: While the use of geographic proxies (e.g., "rich vs. poor" regions or the Area Deprivation Index) effectively illustrates the core arguments, the framework currently leans heavily toward spatial dimensions. Expanding the discussion to include more general, non-spatial structural injustices would further broaden the applicability of the proposed paradigm.

**Support:**

3

---

> ### Author Rebuttal · Authors · 2026-03-31
>
> We are very grateful for the insightful questions and thoughtful comments, as well as the time and effort devoted! Below please see our point-by-point responses:
>
> ---
>
> ### **Q1: "Could you provide a more nuanced clarification of Definition 2.2, specifically regarding the complex interplay and the conceptual boundary between "Social Determinants" and "Sensitive Attributes" in highly entangled data generating processes? Illustrating this distinction with a concrete, real-world scenario would significantly enhance the intuitiveness and clarity of the proposed definition."**
>
> **A1:** Thank you for the helpful suggestion. We provide the following nuanced and formal definition. A social determinant is a variable $S$ that satisfies:
> 1. (Context-level definition) $S$ is defined at the level of a context (e.g., a neighborhood, institution, jurisdiction, or policy environment) rather than as an individual attribute. Multiple individuals share the same value of $S$ by virtue of being situated in the same context.
> 1. (Social-structural content) $S$ characterizes a condition whose cross-context variation is substantially shaped by social-structural forces, such as resource allocation, institutional policy, or systematic investment or disinvestment. This encompasses both directly social conditions (e.g., school funding) and physical or environmental conditions that are socially patterned (e.g., pollution exposure).
> 1. (Exogenous stratification) When $S$ is computed by aggregation over individuals, the grouping over which the aggregation is performed (e.g., a neighborhood boundary, jurisdiction, or institutional membership) is exogenously defined, not derived from the characteristics of the individuals being described.
>
> Consider a simplified but realistic data generating process. Historical redlining channeled Black households into specific neighborhoods, entangling race, zip code, and neighborhood composition through a shared structural history. As a result, these variables remain strongly correlated over time. Despite this entanglement, the definition separates them cleanly:
>
> | Variable | (C1) context-level definition | (C2) social-structural content | (C3) exogenous stratification | Classification |
> |---|---|---|---|---|
> | Applicant's race | **No** | -- | -- | Sensitive attribute |
> | Zip code | Yes | **No** | Yes | Not a social determinant |
> | Racial composition (w.r.t. HOLC-redlined district) | Yes | Yes | **No** | Proxy for sensitive attribute |
> | Racial composition (w.r.t. zip code area) | Yes | Yes | Yes | Social determinant |
> | School funding per pupil (zip code area) | Yes | Yes| Yes | Social determinant |
>
> - `Applicant's race` fails C1.
> - `Zip code` fails C2 since it is an administrative label rather than a measure of any substantive condition. However, it is a proxy for social determinants, since it indexes real structural conditions (school funding, air quality, policing intensity) by partitioning space. This distinction matters for fairness implications: one cannot improve a zip code, but one can improve the school funding or air quality within it.
> - `Racial composition (w.r.t. HOLC-redlined district)` vs. `racial composition (w.r.t. zip code area)`: they differ in their grouping basis. HOLC redlining maps drew district boundaries explicitly based on the racial makeup of residents, making the aggregation endogenous to the characteristic being measured (fails C3), whereas zip code boundaries are postal routes drawn independently of any demographic characteristic (passes C3).
>
> Please let us know if you would have suggestions on how we could further enhance the intuitiveness and clarity.
>
> ---
>
> ### **Q2: "Could the authors elaborate on how the proposed framework can be generalized to quantify non-spatial structural injustices, moving beyond the geographic proxies utilized in the current examples?"**
>
> **A2:** We thank the reviewer for this question. The proposed framework naturally extends beyond geographic proxies: the notion of context can include institutions, jurisdictions, and policy environments, in addition to spatial units. For example:
>
> - Healthcare context: Specialist wait time is defined at the hospital-network level (C1), shaped by reimbursement policy and resource allocation (C2), and aggregated over network membership boundaries (C3).
> - Criminal justice context: Public defender caseload is defined at the jurisdiction level (C1), shaped by legislative funding choices (C2), and aggregated over jurisdictional boundaries (C3).
> - Workplace context: Parental leave policy is defined at the firm level (C1), shaped by corporate and regulatory decisions (C2), and aggregated over organizational boundaries not derived from employee characteristics (C3).

---

> > ### Author Rebuttal · Reviewer_6yWC · 2026-04-02
> >
> > Thanks for the feedback, I maintain my positive score.

---

> > > ### Author Response · Authors · 2026-04-03
> > >
> > > Thank you for getting back to us and for confirming that we have fully addressed your questions and concerns.
> > >
> > > We sincerely appreciate the time and effort you devoted to the review process. Your constructive feedback has helped us further strengthen the manuscript. Thank you again.

---

### Official Review · Reviewer_CFFo · 2026-03-04

**Significance:** 4
**Argument Clarity:** 3
**Rating:** 5
**Confidence:** 3

**Questions:**

1. How can researchers identify the relevant social determinants for a given task?
2. What is a mathematical formulation for the Social determinant parity counterpart described in Pillar 2?
3. Lines 296-298 mention that Figure 1 shows the income distribution is more skewed towards lower income level for PUMAs with higher ADI levels. It is difficult to visualize this difference in the figure. Can the authors quantify this difference in the text?
4. What are the policy implications of the counter-intuitive paradox established in Theorem 4.5? Does it mean that sensitive-attribute-centered mitigation should not be too aggressive?

**Alternative Views Section:**

Yes

**Compliance With Llm Reviewing Policy A Conservative:**

Affirmed.

**Discussion Potential:**

4

**Final Justification:**

I think it is an interesting paper that will garner discussion from the community. The rebuttal has addressed most of the concerns I had so I recommend acceptance.

**Paper Summary:**

This position paper states that the field of machine learning fairness is currently too narrow, focusing on discrimination or bias based on sensitive attributes that fails to capture structural injustice which is often treated as noise rather than signal. The authors argue that ML fairness should also consider this type of injustice by quantifying it through social determinants, which are contextual or environmental factors such as geographic regions or institutional access. The authors identify limitations of the current technical paradigms to capture such factors in terms of data practices, attribute stability, and causal modeling. To support their position, the authors provide a theoretical analysis of college admissions and show that the current quota models focused on sensitive attributes can introduce new structural injustices. Further, they provide a couple of empirical analyses to show that individuals with similar sensitive attributes can have different outcomes based on contextual factors. Finally, the authors propose a research agenda to address structural injustices in ML fairness through actionable pillars.

**Position:**

Yes

**Position In Title:**

Yes

**Related Work:**

3

**Strengths And Weaknesses:**

Strengths
1. The position is defined well and addresses an important conceptual limitation in ML fairness research
2. The authors provide theoretical and empirical evidence to support their position
3. The actionable pillars presented at the end seem reasonable to address the structural injustices highlighted in the paper
4. The paper is well written and relatively easy to follow

Weaknesses
1. The empirical analyses remains largely descriptive. Since the paper critiques ML fairness paradigms, the authors should empirically demonstrate the fairness gap using an ML classifier. The current evaluation does not evaluate an ML system like a classifier or risk prediction model to show how standard fairness metrics fails when contextual variables are ignored. For example, training a classifier on the census data to predict income or on the patient records to predict cancer risk and then comparing fairness metrics with and without social determinant features.
2. The stated position relies on the availability of datasets with granular contextual data, which are often not available, especially in public datasets.
3. There is a practical challenge of determining which social determinants are relevant for a given task and while the authors provide some examples in the appendix, it seems a underspecified. Providing some ways to identify relevant social determinants will help in addressing this.
4. Contextual variables like geographic region can increase privacy and re-identification risks which is concerning, especially in a clinical setting. Even though the paper mentions data governance and differential privacy, there should be some discussion included on how these social determinants could be potentially misused by bad actors.
5. Section 5.1 presents analysis for African American women but section 5.2 switches the analysis to White women. It would be good to extend the analyses to the missing races as well to highlight how social determinants impact vary across different sensitive attributes.

**Support:**

3

---

> ### Author Rebuttal · Authors · 2026-03-31
>
> We are very grateful for the thoughtful questions and constructive feedback! Below please see our point-by-point responses:
>
> ---
>
> ### **Q1: "How can researchers identify the relevant social determinants for a given task?"**
>
> **A1:** We thank the reviewer for this question. We believe identifying the relevant social determinants for a given task requires a combination of domain knowledge, existing literature, and community engagement.
>
> As a starting point, many applied fields maintain well-established frameworks cataloguing structural determinants of outcomes. For example, in public health, the World Health Organization provides a widely used taxonomy [1,3]. In education, factors such as school funding, neighborhood poverty, and access to early childhood programs are well documented as drivers of attainment disparities [2]. Researchers can begin with these existing domain-specific frameworks and narrow based on the specific task and context.
>
> In practice, this process also involves operationalizing these concepts into measurable, context-level variables (e.g., census tract indicators, institutional characteristics), and validating that they capture shared structural conditions rather than individual-level proxies. When multiple plausible determinants exist, reporting results across them can help assess robustness.
>
> Overall, relevance is task-dependent, but the identification of relevant social determinants should be guided by whether a variable captures a shared context that plausibly shapes outcomes, rather than an attribute of the individual.
>
> ---
>
> ### **Q2: "What is a mathematical formulation for the Social determinant parity counterpart described in Pillar 2?"**
>
> **A2:** The Social Determinant Parity we mentioned in Pillar 2 is a direct analogue of Demographic Parity, with stratification defined over a social determinant rather than a sensitive attribute.
>
> Let $S$ denote a discrete social determinant (e.g., percentiles of Area Deprivation Index, ADI), and $\widehat{Y}$ the model prediction. We can define a metric for Social Determinant Parity:
>
> $$
> \max_{s, s'} ~ \Big| \mathbb{E}[\widehat{Y} \mid S = s] - \mathbb{E}[\widehat{Y} \mid S = s'] \Big|.
> $$
>
> ---
>
> ### **Q3: "Lines 296-298 mention that Figure 1 shows the income distribution is more skewed towards lower income level for PUMAs with higher ADI levels. It is difficult to visualize this difference in the figure. Can the authors quantify this difference in the text?"**
>
>
> **A3:** Thank you for the helpful suggestion. We provide quantitative summaries of the income distribution across ADI levels to make this comparison explicit.
>
> Specifically, we report statistics for African American, White, and Asian populations, which illustrate the shift toward lower income levels in higher-ADI regions.
>
> | Region | Race | 25th Percentile | 50th Percentile | 75th Percentile | Mean |
> |:-|:-|-:|-:|-:|-:|
> | Low ADI | African American | $12,000 | $38,000 | $85,000 | $62,395 |
> | Low ADI | White | $12,000 | $50,000 | $108,680 | $85,146 |
> | Low ADI | Asian | $7,000 | $45,000 | $118,000 | $81,837 |
> | Mid ADI | African American | $7,200 | $23,800 | $59,500 | $38,715 |
> | Mid ADI | White | $7,200 | $28,900 | $62,000 | $45,344 |
> | Mid ADI | Asian | $2,200 | $20,000 | $58,800 | $39,561 |
> | High ADI | African American | $6,000 | $18,800 | $42,840 | $33,364 |
> | High ADI | White | $6,000 | $22,000 | $48,000 | $35,030 |
> | High ADI | Asian | $1,000 | $14,000 | $36,320 | $29,105 |
>
>
> ---
>
> ### **Q4: "What are the policy implications of the counter-intuitive paradox established in Theorem 4.5? Does it mean that sensitive-attribute-centered mitigation should not be too aggressive?"**
>
> **A4:** The policy implication of Theorem 4.5 is not that sensitive-attribute-centered mitigation should be less aggressive, but that focusing on sensitive attributes alone provides only a partial view of the problem.
>
> Accordingly, fairness metrics and mitigation strategies should be designed with explicit awareness of the structural conditions shaping outcomes, in addition to the decision-making that immediately produces such outcomes. Theorem 4.5 does not argue for weakening mitigation, but for complementing attribute-based policies with mechanisms that address structural injustice. In particular, policy should attend directly to resource access across regions, rather than relying solely on sensitive-attribute-based mitigation at the point of admission decision-making. More broadly, the result cautions against treating attribute-based mitigation as a self-contained solution, and instead points to the need for a more integrated approach to fairness.
>
> ---
>
> ### References
>
> [1] Orielle Solar, and Alec Irwin. A conceptual framework for action on the social determinants of health. WHO Document Production Services. 2010.
>
> [2] U.S. Department of Health and Human Services. Health People 2030. 2020.
>
> [3] World Health Organization. World report on social determinants of health equity. 2025.

---

> > ### Author Rebuttal · Reviewer_CFFo · 2026-04-01
> >
> > I thank the authors for the responses to my questions, I think they have been answered satisfactorily. However, I don't see any response to the weaknesses I mentioned. What are the authors' thoughts about them?
> >
> > Edit: I thank the reviewers for addressing the weaknesses, especially the empirical analysis which has provided additional support for this position. I will raise my score accordingly.

---

> > > ### Author Response · Authors · 2026-04-03
> > >
> > > Thank you for engaging further and giving us the opportunity to expand our response.
> > >
> > > ---
> > >
> > > ### **W1: "empirically demonstrate the fairness gap using an ML classifier [..., e.g., by] training a classifier [..., and then] comparing fairness metrics with and without social determinant features"**
> > >
> > > **R1:** We thank the reviewer for this suggestion. We include an empirical analysis on the census income prediction task, under a 2$\times$2 design: whether demographic parity (DP) with respect to race is enforced, and whether a social determinant (ADI) is included as a feature. We consider three model classes.
> > >
> > > #### **Table 1: Effect of DP Enforcement on Accuracy and Racial Disparity**
> > > |Predictor|Accuracy (baseline)|Accuracy (DP enforced)|DP metric (baseline)|DP metric (DP enforced)|
> > > |-|-|-|-|-|
> > > |Logistic Regression (w/out ADI)|0.785|0.742|0.130|0.012|
> > > |Logistic Regression (w/ ADI)|0.784|0.736|0.130|0.017|
> > > |Random Forest (w/out ADI)|0.806|0.804|0.115|0.025|
> > > |Random Forest (w/ ADI)|0.806|0.807|0.120|0.014|
> > > |XGBoost (w/out ADI)|0.814|0.810|0.140|0.025|
> > > |XGBoost (w/ ADI)|0.817|0.814|0.148|0.025|
> > > #### **Table 2: Racial Disparity Stratified by Social Determinant After DP Enforcement**
> > > |Predictor|ADI Region|African American|White|Asian|DP metric|
> > > |-|-|-|-|-|-|
> > > |LR (w/out ADI)|Low|0.697|0.693|0.690|0.008|
> > > ||Mid|0.642|0.620|0.583|0.059|
> > > ||High|0.545|0.566|0.455|0.111|
> > > |LR (w/ ADI)|Low|0.762|0.722|0.734|0.040|
> > > ||Mid|0.652|0.648|0.594|0.058|
> > > ||High|0.463|0.540|0.364|0.176|
> > > |RF (w/out ADI)|Low|0.651|0.713|0.693|0.062|
> > > ||Mid|0.622|0.615|0.552|0.070|
> > > ||High|0.545|0.549|0.455|0.094|
> > > |RF (w/ ADI)|Low|0.679|0.730|0.709|0.051|
> > > ||Mid|0.630|0.602|0.552|0.077|
> > > ||High|0.480|0.490|0.443|0.048|
> > > |XGB (w/out ADI)|Low|0.725|0.698|0.681|0.044|
> > > ||Mid|0.625|0.600|0.536|0.089|
> > > ||High|0.504|0.517|0.419|0.098|
> > > |XGB (w/ ADI)|Low|0.688|0.730|0.708|0.041|
> > > ||Mid|0.627|0.594|0.512|0.116|
> > > ||High|0.504|0.477|0.376|0.129|
> > >
> > > We find that:
> > > 1. including ADI as a feature does not consistently improve DP, and can sometimes worsen observed DP violations;
> > > 1. when outcomes are stratified by social determinant, disparities become more pronounced.
> > >
> > > Both findings are expected and support our position: standard sensitive-attribute-based fairness metrics effectively smooth over structural contexts, thereby obscuring variation across them. Moreover, including social determinants as predictive features, even when appropriate, is not equivalent to treating them as an evaluative axis for fairness.
> > >
> > > ---
> > >
> > > ### **W2: "The stated position relies on the availability of datasets with granular contextual data, which are often not available, especially in public datasets."**
> > >
> > > **R2:** We agree that granular contextual variables are often unavailable in existing (especially public) datasets, and may even be excluded during data processing despite their relevance to structural context (Section 3.1). Our framework does not assume their availability, but instead emphasizes the need to actively seek out latent factors that could reflect structural characteristic (Section 7).
> > >
> > > ---
> > >
> > > ### **W3: "Providing some ways to identify relevant social determinants will help in addressing [practical challenge of determining which ones are relevant for a given task]."**
> > >
> > > **R3:** We thank reviewer for this suggestion. We believe this is closely related to **Q1**, and we will include **A1** in our revised manuscript.
> > >
> > > ---
> > >
> > > ### **W4: "Even though the paper mentions data governance and differential privacy, there should be some discussion included on how these social determinants could be potentially misused by bad actors."**
> > >
> > > **R4:** We will include an explicit discussion of potential misuse risks. For instance,
> > > 1. Re-identification via context linkage: cross-referencing granular context units with external geospatial or administrative records to identify individuals, particularly when context membership is sparse.
> > > 1. Proxy laundering (individual-/context- level): deliberately adopting a social-determinant related variable as a legally permissible proxy for protected attributes (individual-level) or neighborhood (context-level), exploiting the context-level framing to enact discrimination while claiming neutrality.
> > > 1. Strategic manipulation of context: leveraging influence over context boundaries or policy (e.g., redistricting, institutional reclassification) to alter the value of social determinants and shift model outputs.
> > >
> > > ---
> > >
> > > ### **W5: "It would be good to extend the analyses to the missing races as well to highlight how social determinants impact vary across different sensitive attributes."**
> > >
> > > **R5:** We thank reviewer for this suggestion. We believe this is closely related to **Q3** and **W1**. In light of your suggestion, we will include the analyses and results in **A3** and **R1** in our revised manuscript to more concretely and transparently illustrate how the impact of social determinants varies across sensitive attributes.
> > >
> > > ---
> > >
> > > Thank you again for the constructive and thoughtful feedback.

---

### Official Review · Reviewer_jnPS · 2026-03-09

**Significance:** 4
**Argument Clarity:** 3
**Rating:** 5
**Confidence:** 3

**Questions:**

1) What are your minimum criteria for labeling a feature as a “social determinant” (vs a sensitive proxy), and what safeguards do you recommend to prevent misuse?

2) In the high-stakes example(s), what analyses support that observed disparities reflect structural barriers rather than unobserved confounding or measurement artifacts, and how should practitioners communicate this uncertainty?

**Alternative Views Section:**

Yes

**Compliance With Llm Reviewing Policy A Conservative:**

Affirmed.

**Discussion Potential:**

4

**Paper Summary:**

The paper argues that fairness evaluations focused primarily on sensitive attributes can miss structural drivers of inequity (e.g., geography, access, socioeconomic conditions) and therefore should incorporate social determinants into fairness measurement. It advocates auditing and interpreting disparities through these determinants (including within-group disparities where sensitive attributes are fixed), and illustrates the argument with conceptual and empirical examples (including a high-stakes healthcare screening setting) to show how structural factors can shape outcomes and fairness conclusions. The main contribution is a call to action and framing for fairness practice that treats social determinants as first-class variables in evaluation and policy discussion.

**Position:**

Yes

**Position In Title:**

Yes

**Related Work:**

3

**Strengths And Weaknesses:**

**Strengths**

* Timely and important position: reframes fairness toward structural inequity, which is highly relevant for real-world deployments.

* Concrete motivating examples (including high-stakes context) make the argument intuitive and discussion-worthy.

* Likely to spark productive debate on what fairness should measure (and who is responsible for which interventions).

**Weaknesses**

* Needs sharper operational guidance: what qualifies as a “social determinant” vs a proxy for sensitive attributes, and how to avoid “proxy laundering.”

* Would probalby benefit from a more explicit “how-to” (e.g., a minimal checklist for reviewers/practitioners: variable selection, granularity, intersectional reporting, robustness).

* More clarity on the boundary between ML interventions vs policy interventions, and what claims are appropriate for each.

**Support:**

3

---

> ### Author Rebuttal · Authors · 2026-03-31
>
> We are very grateful for the constructive feedback and insightful questions, as well as the time and effort devoted! Below please see our point-by-point responses:
>
> ---
>
> ### **Q1: "What are your minimum criteria for labeling a feature as a 'social determinant' (vs a sensitive proxy), and what safeguards do you recommend to prevent misuse?"**
>
> **A1:** We use the following definition to specify minimum criteria: a social determinant is a variable $S$ that satisfies
> 1. (Context-level definition) $S$ is defined at the level of a context (e.g., a neighborhood, institution, jurisdiction, or policy environment) rather than as an individual attribute. Multiple individuals share the same value of $S$ by virtue of being situated in the same context.
> 1. (Social-structural content) $S$ characterizes a condition whose cross-context variation is substantially shaped by social-structural forces, such as resource allocation, institutional policy, or systematic investment or disinvestment. This encompasses both directly social conditions (e.g., school funding) and physical or environmental conditions that are socially patterned (e.g., pollution exposure).
> 1. (Exogenous stratification) When $S$ is computed by aggregation over individuals, the grouping over which the aggregation is performed (e.g., a neighborhood boundary, jurisdiction, or institutional membership) is exogenously defined, not derived from the characteristics of the individuals being described.
>
> We recommend safeguards along three pillars (Section 7) to prevent misuse (e.g., redlining-style practices, or proxy laundering):
> (i) Data governance: use privacy-aware, aggregated, and well-defined contextual variables with tiered access, avoiding individual-level inference or reconstruction;
> (ii) Metric design: employ social determinants as measurement instruments rather than default constraints to enforce during model design;
> (iii) Data generating process analysis: use disparities across social determinants to identify structural intervention levers rather than additional axes for individual-level discrimination.
>
> > Please also see the **practical mapping table** provided in our response to Reviewer `6yWC`, which demonstrates how these criteria distinguish social determinants in realistic, entangled scenarios.
>
> ---
>
> ### **Q2: "In the high-stakes example(s), what analyses support that observed disparities reflect structural barriers rather than unobserved confounding or measurement artifacts, and how should practitioners communicate this uncertainty?"**
>
> **A2:** We appreciate this question, which resonates with a core motivation of our work. We want to be candid that our semi-synthetic simulation (as noted in Footnote 7) abstracts from the full complexity of the underlying epidemiological processes, and we do not intend to make causal identification claims. The data and available tools in our setting do not support such analysis, and our focus is instead on isolating mechanisms under controlled conditions.
>
> On communicating uncertainty, we agree this is an important direction. We suggest that practitioners present such results as evidence of structurally aligned disparities, while explicitly acknowledging alternative explanations (e.g., confounding or measurement limitations) if the technical tools do not come with identification guarantee. At the same time, unresolved uncertainty about the sources of disparities should not be taken as evidence of neutrality in the baseline conditions. We would be happy to incorporate further discussion. Please let us know if you have suggestions on how we could further strengthen this aspect.

---

> > ### Author Rebuttal · Reviewer_jnPS · 2026-04-02
> >
> > The rebuttal usefully clarifies the definition of “social determinants,” discusses safeguards against misuse (including proxy laundering), and better scopes the healthcare example as primarily descriptive/illustrative rather than making strong causal identification claims. This addresses part of my original concerns.
> >
> > However, the paper still lacks actionable operational guidance: a reviewer-/practitioner-facing “minimum viable” audit protocol specifying how to select/granularize social determinants, what robustness/sensitivity checks to run, and how to prevent proxies from reintroducing sensitive-attribute discrimination. Clearer, concrete guidance is important given the paper’s central call to quantify structural injustice via social determinants.
> >
> > Follow-up questions :
> >
> > 1) Could you add a short “Minimum Viable Audit Protocol” (5–8 bullets) that reviewers/practitioners can apply consistently, including required slice reporting, recommended robustness checks, and guidance on granularity/aggregation of social determinants?
> >
> > 2) How do you recommend distinguishing a legitimate social determinant from a sensitive proxy in practice? Please provide concrete safeguards (e.g., tests/diagnostics, governance constraints, or documentation requirements) to reduce “proxy laundering.”
> >
> > 3) For the healthcare example, can you add an explicit Scope of Claims statement clarifying what is descriptive vs causal, and what additional evidence would be needed to support stronger “structural barrier” interpretations (e.g., sensitivity analyses, alternative stratifications, or external validation)?

---

> > > ### Author Response · Authors · 2026-04-03
> > >
> > > Thank you for getting back to us, and for the opportunity to provide further response and clarification.
> > >
> > > ---
> > >
> > > ### **Q3: "Could you add a short 'Minimum Viable Audit Protocol' (5–8 bullets) that reviewers/practitioners can apply consistently, including required slice reporting, recommended robustness checks, and guidance on granularity/aggregation of social determinants?"**
> > >
> > > **A3:** We operationalize our definition of social determinants into a minimal viable audit protocol:
> > >
> > > 1. **Validate context-level definition**
> > >    - Confirm that each $S$ variable is defined at a context level (e.g., neighborhood, institution, jurisdiction), not as an individual attribute.
> > > 2. **Validate structural grounding of $S$**
> > >    - Provide a concrete mechanism linking variation in $S$ to social-structural processes (e.g., policy rules, institutional practices, resource allocation).
> > > 3. **Check exogenous stratification**
> > >    - Ensure that any aggregation used to compute $S$ relies on externally defined groupings (e.g., geographic or institutional boundaries), not clusters derived from individual features.
> > > 4. **Set and audit granularity**
> > >    - Re-evaluate results under coarser and finer versions of the same context definition (e.g., following the spirit of geographic aggregation sensitivity and de-identification analyses in census disclosure review).
> > > 5. **Slice reporting by $S$**
> > >    - Report key metrics (e.g., performance, empirical violations of standard sensitive-attribute-based fairness) stratified by $S$-defined contexts.
> > > 6. **Proxy correlation audit and mitigation**
> > >    - Quantify correlations between $S$ and sensitive attributes at the context level.
> > >    - If correlations exceed a threshold, either (i) coarsen $S$, (ii) remove or replace the variable, or (iii) explicitly justify its use and qualify downstream interpretations.
> > > 7. **Robustness to specification**
> > >    - Recompute $S$ under alternative plausible context definitions (e.g., different boundaries or institutional groupings).
> > >    - Verify that substantive conclusions remain stable across these specifications.
> > > 8. **Feedback loop and deployment monitoring**
> > >    - Track whether decisions informed by the model induce systematic changes in the distribution of $S$ across contexts.
> > >
> > > We will include this list in our revised manuscript.
> > >
> > > ---
> > >
> > > ### **Q4: "How do you recommend distinguishing a legitimate social determinant from a sensitive proxy in practice? Please provide concrete safeguards (e.g., tests/diagnostics, governance constraints, or documentation requirements) to reduce 'proxy laundering.'"**
> > >
> > > **A4:** Thank reviewer for this question. In practice, we can distinguish legitimate social determinants $S$ from proxy laundering through complementary safeguards:
> > >
> > > - **Tests / diagnostics**
> > >   1. *Proxy correlation audit:* Measure correlation between $S$ and sensitive attributes at the context level. Treat high correlation as a risk signal.
> > >   1. *Boundary robustness:* Recompute $S$ under alternative context definitions. Instability suggests incidental correlations rather than structural grounding.
> > > - **Governance constraints**
> > >   1. *Exogeneity requirement:* Restrict $S$ to variables defined over externally specified contexts, and exclude groupings derived from individual features.
> > >   1. *Usage constraint:* Limit $S$ to context-level auditing and prohibit individual-level decision-making use (unless a clear, documented justification is provided).
> > >
> > > - **Documentation requirements**
> > >   - For each $S$, document: (i) construction and data sources, (ii) structural mechanism linking $S$ to social-structural processes, (iii) observed correlations with sensitive attributes, and (iv) any mitigation steps taken.
> > >
> > > We will include this discussion in our revised manuscript.
> > >
> > > ---
> > >
> > > ### **Q5: "For the healthcare example, can you add an explicit Scope of Claims statement clarifying what is descriptive vs causal, and what additional evidence would be needed to support stronger 'structural barrier' interpretations (e.g., sensitivity analyses, alternative stratifications, or external validation)?"**
> > >
> > > **A5:** We thank the reviewer for this suggestion. We will make this distinction explicit in the experimental section.
> > >
> > > Specifically, we will clarify that our semi-synthetic analysis in the healthcare part is descriptive. We do not intend to make, nor should our results be interpreted as, causal claims about structural barriers. We will also briefly outline what additional evidence would be required to support stronger interpretations, such as causal representation learning with identification guarantees, or consistent replication across contexts and alternative stratifications.
> > >
> > > ---
> > >
> > > Thank you again for your continued engagement and constructive feedback, as well as the time and effort you have devoted. Your suggestions have helped make our paper clearer and more transparent.

---

### Official Review · Reviewer_EELW · 2026-03-16

**Significance:** 3
**Argument Clarity:** 3
**Rating:** 4
**Confidence:** 4

**Questions:**

1. Can you provide a more concrete formal definition of what counts as a social determinant in your framework, especially in borderline cases where a variable may be highly correlated with protected attributes or may itself act like an individual-level proxy?
2. You propose moving fairness assessment toward social-determinant-aware metrics such as Social Determinant Parity. What is the simplest precise version of such a metric that you believe should be evaluated today, and how should it be compared against standard group fairness or causal fairness criteria in practice?
3. The theoretical admissions example is compelling, but it is also stylized and abstracts away other forms of discrimination. Which assumptions are essential for the paradox to hold, and which can be relaxed without changing the conclusion?

**Alternative Views Section:**

Yes

**Compliance With Llm Reviewing Policy A Conservative:**

Affirmed.

**Discussion Potential:**

3

**Paper Summary:**

This position paper argues that machine learning fairness should move beyond viewing unfairness mainly as discrimination over sensitive attributes and should instead quantify structural injustice through social determinants, which the authors define as contextual features of the data-generating process, such as neighborhood conditions, environmental exposures, and access-related factors that shape opportunities and outcomes without being attributes of a specific individual.The paper’s contributions are threefold. First, it offers a conceptual and cross-disciplinary framing that contrasts sensitive attributes with social determinants and situates the latter within political philosophy, sociology, economics, and healthcare. Second, it argues that current fairness paradigms are limited because common datasets and benchmarks often omit contextual variables, standard fairness metrics centered on stable protected attributes miss within-group heterogeneity induced by context, and prevailing causal fairness models are largely individual-level rather than community-level. Third, it supports this position with a theoretical college-admissions analysis showing that mitigation focused only on sensitive attributes can create new structural injustice, along with empirical illustrations from U.S. census data and a breast-cancer screening case study showing that people with the same sensitive attributes can experience materially different outcomes across different social contexts

**Position:**

Yes

**Position In Title:**

Yes

**Related Work:**

3

**Strengths And Weaknesses:**

Strengths:

1. The paper presents a clear and distinctive position from the title through the conclusion, so the central claim is easy to identify and evaluate.
2. It supports that position with several kinds of evidence, including conceptual framing, a theoretical admissions example, and empirical illustrations from census and breast cancer screening settings.
3. The topic is highly relevant to ICML because it speaks directly to fairness benchmarks, evaluation, and causal modeling, and the Alternative Views section makes it likely to spark discussion.

Weaknesses:

1. The core idea is still not operationalized enough for practice, so the paper is stronger as an agenda setting piece than as a concrete methodological blueprint.
2. The empirical evidence is suggestive rather than decisive, since it shows contextual heterogeneity but does not fully establish how the proposed direction should be implemented and validated against existing fairness approaches.
3. Some claims about the limits of current fairness methods feel broader than the evidence shown, especially when the paper moves from showing incompleteness in common practice to implying deeper mismatch at the paradigm level.

**Support:**

3

---

> ### Author Rebuttal · Authors · 2026-03-31
>
> We are very grateful for the insightful questions and constructive comments! Below please see our point-by-point response:
>
> ---
>
> ### **Q1: "Can you provide a more concrete formal definition of what counts as a social determinant in your framework, especially in borderline cases where a variable may be highly correlated with protected attributes or may itself act like an individual-level proxy?"**
>
> **A1:** We provide the following concrete formal definition. A social determinant is a variable $S$ that satisfies:
> 1. (Context-level definition) $S$ is defined at the level of a context (e.g., a neighborhood, institution, jurisdiction, or policy environment) rather than as an individual attribute. Multiple individuals share the same value of $S$ by virtue of being situated in the same context.
> 1. (Social-structural content) $S$ characterizes a condition whose cross-context variation is substantially shaped by social-structural forces, such as resource allocation, institutional policy, or systematic investment or disinvestment. This encompasses both directly social conditions (e.g., school funding) and physical or environmental conditions that are socially patterned (e.g., pollution exposure).
> 1. (Exogenous stratification) When $S$ is computed by aggregation over individuals, the grouping over which the aggregation is performed (e.g., a neighborhood boundary, jurisdiction, or institutional membership) is exogenously defined, not derived from the characteristics of the individuals being described.
>
> Correlation with protected attributes does not disqualify a variable. For instance, neighborhood poverty rate may correlate with race precisely because it reflects underlying structural inequities that the notion is intended to capture.
>
> Criterion 3 guards against relabeling an individual-level proxy as a social determinant, because the grouping over which any aggregation is performed must correspond to a pre-specified contextual unit (e.g., a census tract, school district, or policy jurisdiction). In particular, singling out a person as a degenerate "aggregation" (as in individual-level proxy) does not quality.
>
> > Please also see the **practical mapping table** provided in our response to Reviewer `6yWC`, which demonstrates how these criteria distinguish social determinants in realistic, entangled scenarios.
>
> ---
>
> ### **Q2: "What is the simplest precise version of [Social Determinant Parity] metric that you believe should be evaluated today, and how should it be compared against standard group fairness or causal fairness criteria in practice?"**
>
> **A2:** A simplest and precise instantiation of Social Determinant Parity is a direct analogue of Demographic Parity, with stratification defined over a social determinant rather than a sensitive attribute.
>
> Formally, let $S$ denote a discrete social determinant (e.g., percentiles of Area Deprivation Index, ADI), and $\widehat{Y}$ the model prediction, and we can define a metric for Social Determinant Parity:
>
> $$
> \max_{s, s'} ~ \Big| \mathbb{E}[\widehat{Y} \mid S = s] - \mathbb{E}[\widehat{Y} \mid S = s'] \Big|.
> $$
>
> In practice, **we compare these notions by what they measure, not by whether they agree,** and the comparison consists of evaluating all criteria on the same model and interpreting each relative to its stratification. Standard group fairness and causal fairness metrics capture disparity centered around sensitive attributes, while social determinant–based metrics reveal how model outputs distribute benefits and burdens across structural contexts. The goal is not compatibility or substitution, but a more complete evaluation in which structurally induced disparities can be detected beyond attribute-based ones.
>
> ---
>
> ### **Q3: "Which assumptions are essential for the paradox to hold [in Theorem 4.5], and which can be relaxed without changing the conclusion?"**
>
> **A3:** Assumptions 4.1--4.4 make the setting explicit while keeping the mechanism minimal:
> - (Rather mild assumptions) Region-specific demographic composition (Assumption 4.1) and limited admission capacity (Assumption 4.4) are required to define the allocation setting (heterogeneous populations and opportunity scarcity). Stochastic dominance of preparedness across regions (Assumption 4.3) captures a directional disparity, without depending on a specific distribution family.
> - (Relatively stronger assumption) Preparedness determined at the regional level (Assumption 4.2) is a simplifying assumption that isolates structural disparities from intra-region mechanisms, without restricting the generality of the conclusion.
>
> Our conclusion does not require Assumption 4.2 in its strict form. Relaxing it to allow intra-region heterogeneity or additional discrimination mechanisms preserves the effect, because the core mechanism of the paradox arises from the interaction between attribute-based mitigation and cross-region disparities under capacity constraints.

---

> > ### Author Rebuttal · Reviewer_EELW · 2026-04-04
> >
> > Thanks for the response. I do not have further questions.

---

> > > ### Author Response · Authors · 2026-04-04
> > >
> > > Thank you for getting back to us and for confirming that concerns/questions have been fully addressed.
> > >
> > > We are very grateful for the thoughtful questions and constructive feedback. Thanks again.

---

### Decision · Program_Chairs · 2026-04-30

**Decision:**

Accept (regular)

**Comment:**

Acceptance was recommended based on the consensus.